# Shared neural underpinnings of multisensory integration and trial-by-trial perceptual recalibration in humans

**Hame Park[1,2,3]\*, Christoph Kayser[1,2]\***

[1]Department for Cognitive Neuroscience, Faculty of Biology, Bielefeld University, Bielefeld, Germany; [2]Center of Excellence Cognitive Interaction Technology, Bielefeld University, Bielefeld, Germany; [3]Institute of Neuroscience and Psychology, University of Glasgow, Glasgow, United Kingdom

**Abstract** Perception adapts to mismatching multisensory information, both when different cues appear simultaneously and when they appear sequentially. While both multisensory integration and adaptive trial-by-trial recalibration are central for behavior, it remains unknown whether they are mechanistically linked and arise from a common neural substrate. To relate the neural underpinnings of sensory integration and recalibration, we measured whole-brain magnetoencephalography while human participants performed an audio-visual ventriloquist task. Using single-trial multivariate analysis, we localized the perceptually-relevant encoding of multisensory information within and between trials. While we found neural signatures of multisensory integration within temporal and parietal regions, only medial superior parietal activity encoded past and current sensory information and mediated the perceptual recalibration within and between trials. These results highlight a common neural substrate of sensory integration and perceptual recalibration, and reveal a role of medial parietal regions in linking present and previous multisensory evidence to guide adaptive behavior.
DOI: https://doi.org/10.7554/eLife.47001.001

**\*For correspondence:**
hame.park@uni-bielefeld.de (HP);
christoph.kayser@uni-bielefeld.de
(CK)

**Competing interests:** The authors declare that no competing interests exist.

## Introduction

Multisensory information offers substantial benefits for behavior. For example, acoustic and visual cues can be combined to derive a more reliable estimate of where an object is located (*Alais and Burr, 2004*; *Ernst and Banks, 2002*; *Körding et al., 2007*; *Wozny and Shams, 2011b*). Yet, the process of multisensory perception does not end once an object is removed. In fact, multisensory information can be exploited to calibrate subsequent perception in the absence of external feedback (*Frissen et al., 2012*; *Wozny and Shams, 2011a*). In a ventriloquist paradigm, for example, the sight of the puppet and the actor's voice are combined when localizing the speech source, and both cues influence the localization of subsequent unisensory acoustic cues, if probed experimentally (*Bosen et al., 2017*; *Bosen et al., 2018*; *Bruns and Röder, 2015*; *Bruns and Röder, 2017*; *Callan et al., 2015*; *Radeau and Bertelson, 1974*; *Recanzone, 1998*). This trial-by-trial recalibration of perception by previous multisensory information has been demonstrated for spatial cues, temporal cues, and speech signals (*Kilian-Hütten et al., 2011a*; *Lüttke et al., 2016*; *Lüttke et al., 2018*; *Van der Burg et al., 2013*), and has been shown to be modulated by attention (*Eramudugolla et al., 2011*). Despite the importance of both facets of multisensory perception for adaptive behavior - the combination of information within a trial and the trial-by-trial adjustment of perception - it remains unclear whether they originate from shared neural mechanisms.

In fact, the neural underpinnings of trial-by-trial recalibration remain largely unclear. Those studies that have investigated neural correlates of multisensory recalibration mostly focused on the

**eLife digest** A good ventriloquist will make their audience experience an illusion. The speech the spectators hear appears to come from the mouth of the puppet and not from the puppeteer. Moviegoers experience the same illusion: they perceive dialogue as coming from the mouths of the actors on screen, rather than from the loudspeakers mounted on the walls. Known as the ventriloquist effect, this 'trick' exists because the brain assumes that sights and sounds which occur at the same time have the same origin, and it therefore combines the two sets of sensory stimuli.

A version of the ventriloquist effect can be induced in the laboratory. Participants hear a sound while watching a simple visual stimulus (for instance, a circle) appear on a screen. When asked to pinpoint the origin of the noise, volunteers choose a location shifted towards the circle, even if this was not where the sound came from. In addition, this error persists when the visual stimulus is no longer present: if a standard trial is followed by a trial that features a sound but no circle, participants perceive the sound in the second test as 'drawn' towards the direction of the former shift. This is known as the ventriloquist aftereffect.

By scanning the brains of healthy volunteers performing this task, Park and Kayser show that a number of brain areas contribute to the ventriloquist effect. All of these regions help to combine what we see with what we hear, but only one maintains representations of the combined sensory inputs over time. Called the medial superior parietal cortex, this area is unique in contributing to both the ventriloquist effect and its aftereffect.

We must constantly use past and current sensory information to adapt our behavior to the environment. The results by Park and Kayser shed light on the brain structures that underpin our capacity to combine information from several senses, as well as our ability to encode memories. Such knowledge should be useful to explore how we can make flexible decisions.

DOI: https://doi.org/10.7554/eLife.47001.002

adaptation following long-term (that is, often minutes of) exposure to consistent multisensory discrepancies (*Bruns et al., 2011*; *Zierul et al., 2017*). However, we interact with our environment using sequences of actions dealing with different stimuli, and thus systematic sensory discrepancies as required for long-term effects are possibly seldom encountered. Hence, while the behavioral patterns of multisensory trial-by-trial recalibration are frequently studied (*Bosen et al., 2017*; *Bruns and Röder, 2015*; *Delong et al., 2018*; *Van der Burg et al., 2018*; *Wozny and Shams, 2011a*) it remains unclear when and where during sensory processing their neural underpinnings emerge.

In contrast to this, the neural underpinnings of multisensory integration of simultaneously received information have been investigated in many paradigms and model systems (*Angelaki et al., 2009*; *Bizley et al., 2016*; *Fetsch et al., 2013*). Studies on spatial ventriloquist-like paradigms, for example, demonstrate contributions from auditory and parietal cortex (*Bonath et al., 2014*; *Bruns and Röder, 2010*; *Bruns and Röder, 2015*; *Callan et al., 2015*; *Harvey et al., 2014*; *Bonath et al., 2007*; *Starke et al., 2017*). A series of recent studies demonstrates that posterior parietal regions automatically fuse multisensory information, while anterior parietal regions give way to a more flexible spatial representation that follows predictions from Bayesian causal inference (*Cao et al., 2019*; *Rohe et al., 2019*; *Rohe and Noppeney, 2015b*; *Rohe and Noppeney, 2016*). Given that parietal regions also contribute to the maintenance of sensory information within or between trials (*Harvey et al., 2012*; *Morcos and Harvey, 2016*; *Raposo et al., 2014*; *Schott et al., 2018*; *Uncapher and Wagner, 2009*; *Vilberg and Rugg, 2008*) this raises the possibility that parietal regions are in fact mediating both the combination of sensory information within a trial, and the influence of such an integrated representation on guiding subsequent adaptive behavior.

To link the neural mechanisms underlying multisensory integration and trial-by-trial recalibration, we measured whole-brain activity using magnetoencephalography (MEG) while human participants performed a spatial localization task (*Figure 1A*). The paradigm was designed to reveal the behavioral correlates of audio-visual integration (i.e. the ventriloquist effect, VE) and the influence of this on the localization of a subsequent unisensory sound (the ventriloquist aftereffect, VAE) (*Wozny and Shams, 2011a*). Using single-trial classification we determined the relevant neural representations of auditory and visual spatial information and quantified when and where these are influenced by

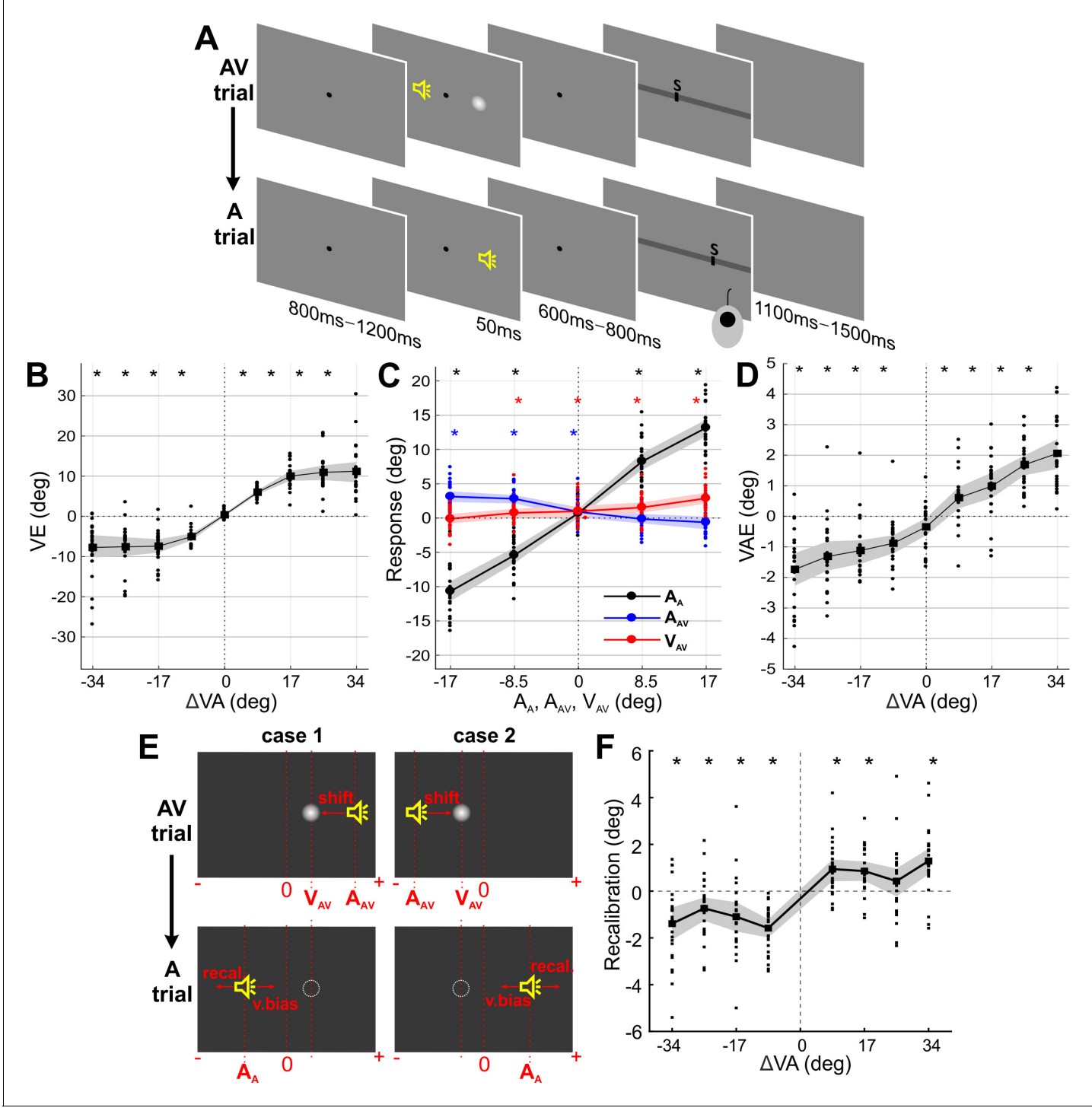

**Figure 1.** Paradigm and behavioral results (N = 24). (**A**) Experimental design. Participants localized auditory (or visual) targets and indicated the perceived location using a mouse cursor. Audio-visual (AV) and auditory (A) trials alternated. (**B**) Response bias induced by the ventriloquist effect (VE) as a function of audio-visual discrepancy in the AV trial. VE: the difference between the reported location ($R_{AV}$) and the location at which the sound ($A_{AV}$) was actually presented ($R_{AV} - A_{AV}$). (**C**) Sound localization response in the A trial was significantly influenced by the current sound ($A_A$; black), the previous sound ($A_{AV}$; blue) and the previous visual ($V_{AV}$; red) stimulus. (**D**) Response bias induced by the ventriloquist effect (VAE) as a function of audio-visual discrepancy in the AV trial. VAE: the difference between the reported location ($R_A$) minus the mean reported location for all trials of the same stimulus position ($R_A$ – mean($R_A$)). (**E**) Example trials dissociating a pure visual bias from a genuine multisensory bias in the VAE. Trials for which the expected visual (v.bias) and multisensory (recal) biases are in opposite directions were selected; these satisfied either case 1: $V_{AV} - A_{AV} < 0$ and $A_A \leq V_{AV}$ or case 2; $V_{AV} - A_{AV} > 0$ and $A_A \geq V_{AV}$. (**F**) Recalibration bias for trials from (**E**). Solid lines indicate mean across participants. Shaded area is the

*Figure 1 continued on next page*

*Figure 1 continued*

estimated 95% confidence interval based on the bootstrap hybrid method. Dots denote individual participants. Asterisks denote p-values<0.05 from two-sided Wilcoxon signed rank tests, corrected with the Holm method for multiple comparisons, $A_A$: sound location in A trial. $A_{AV}$: sound location in AV trial. $V_{AV}$: visual location in AV trial. Deposited data: Data_behav (folder).

DOI: https://doi.org/10.7554/eLife.47001.003

previous sensory evidence. We then modeled the influence of these candidate neural representations on the participant-specific trial-by-trial response biases. As expected based on previous work, our results reveal neural correlates of sensory integration in superior temporal and parietal regions. Importantly, of these, only activity within the superior parietal cortex encodes current multisensory information and retains information from preceding trials, and uses both to guide adaptive behavior within and across trials.

## Results

24 volunteering participants localized sounds in alternating sequences of audio-visual (AV) and auditory (A) trials (*Figure 1A*). During audio-visual trials spatially localized sounds were accompanied by visual stimuli at the same or a different location. Importantly, the AV trial always preceded the A trial. Within and between trials, the positons of auditory and visual stimuli were sampled randomly and semi-independently from five locations ($i = -17°, -8.5°, 0°, 8.5°, 17°$ from the midline ($0°$); see Materials and methods for additional details). Participants fixated a central fixation dot before, during, and after the stimuli, but were free to move their eyes during the response period.

### Behavioral results - Ventriloquist effect

Behavioral responses in AV trials revealed a clear ventriloquist effect (VE) as a function of the presented audio-visual discrepancy ($\Delta VA = V_{AV} - A_{AV}$), whereby the visual stimulus biased the perceived sound location (*Figure 1B*). The VE was computed as the difference between participant's response (R) and the actual sound location for that trial (i.e., $R_{AV} - A_{AV}$, where subscript denotes the trial type). Model comparison revealed that both stimuli had a significant influence on the participants' responses (relative BIC values of three candidate models, c.f. Materials and methods Section: $mi_1$: 938, $mi_2$: 3816, $mi_3$: 0; relative AIC values; $mi_1$: 945, $mi_2$: 3823, $mi_3$: 0; protected exceedance probability (*Rigoux et al., 2014*); $mi_1$: 0, $mi_2$: 0, $mi_3$: 1; winning model: $mi_3$: VE ~ 1 + β·$A_{AV}$ + β·$V_{AV}$), with significant contributions from both the auditory ($A_{AV}$), and visual stimuli ($V_{AV}$) ($\beta_{A\_AV}$ = -0.48, $\beta_{V\_AV}$ = 0.22, $t_{A\_AV}$ = -70.0, $t_{V\_AV}$ = 31.7, $p_{A\_AV}$, $p_{V\_AV}$ <0.01, d.f. = 8064). Across participants, the VE bias was significant for each non-zero audio-visual discrepancy (all $p<10^{-4}$; Wilcoxon signed rank tests, corrected for multiple tests with the Holm procedure).

### Behavioral results - Ventriloquist aftereffect

Participants localized the sound in the A trials reliably (*Figure 1C*, black graph), with the data exhibiting a well-known central bias (*Rohe and Noppeney, 2015a*). This confirms that the convolution with HRTFs indeed led to sounds that were perceived as spatially dispersed. Behavioral responses in A trials revealed a significant ventriloquist aftereffect (VAE; *Figure 1D*) as a function of the audio-visual discrepancy ($\Delta VA$) in the previous AV trial, demonstrating that the preceding multisensory stimuli had a lasting influence on the localization of subsequent sounds. The VAE for each sound location was computed as $R_A - mean(R_A)$; whereby $mean(R_A)$ reflects the mean over all localization responses for this position and participant. This approach ensures that any bias in pure auditory localization does not confound the VAE effect (*Rohe and Noppeney, 2015a*; *Wozny and Shams, 2011a*). Model comparison revealed that both previous stimuli had a significant influence on the VAE (relative BIC values of three candidate models, c.f. Materials and methods section; $mr_1$: 27, $mr_2$: 357, $mr_3$: 0; relative AIC values: $mr_1$: 34, $mr_2$: 364, $mr_3$: 0; protected exceedance probability; $mr_1$: 0, $mr_2$: 0, $mr_3$: 1; winning model: $mr_3$: VAE ~ 1 + β·$A_{AV}$ + β·$V_{AV}$), with significant contributions from the previous sound ($A_{AV}$), and the previous visual stimulus ($V_{AV}$) ($\beta_{A\_AV}$ = -0.09, $\beta_{V\_AV}$ = 0.03, $t_{A\_AV}$ = -19.4, $t_{V\_AV}$ = 6.0, $p_{A\_AV}$, $p_{V\_AV}$ <0.01, d.f. = 8064). Note that because the VAE was defined relative to the average perceived location for each sound position, the actual sound position ($A_A$) does not

contribute to the VAE. Across participants, the VAE bias was significant for each non-zero audio-visual discrepancy (all $p<10^{-2}$; two-sided Wilcoxon signed rank tests, corrected for multiple tests with the Holm procedure).

We performed two control analyses to further elucidate the nature of the VAE. First, we asked whether the shift in the perceived sound location was the result of a bias towards the previous visual stimulus location, or a bias induced specifically by the previous audio-visual discrepancy (*Wozny and Shams, 2011a*). To dissect these hypotheses, we selected trials for which the expected biases arise from the direction of the VE, and not from a visual bias towards $V_{AV}$ (*Figure 1E*). The data were clearly in favor of a genuine multisensory bias, as the VAE remained significant for these trials (all $p<0.05$; except for +25.5 condition; two-sided Wilcoxon signed rank tests, corrected for multiple tests; *Figure 1F*) (*Wozny and Shams, 2011a*). Second, we asked whether the response bias in the A trial was better accounted for by the sensory information in the previous trial (i.e. the previous multisensory discrepancy: $\Delta VA$) or the participant's response in that trial ($R_{AV}$). Formal model comparison revealed that the model $R_A \sim 1 + A_A + \Delta VA$ provided a better account of the data than a response-based model $R_A \sim 1 + A_A + R_{AV}$ (relative BIC: 0, 393; BIC weights: 1, 0), supporting the notion that recalibration is linked more to the physical stimuli than the participants response (*Van der Burg et al., 2018*).

## Representations of single trial sensory information in MEG source data

The analysis of the MEG data was designed to elucidate the neural underpinnings of the VAE and to contrast these to the neural correlates of the VE. Specifically, we first determined neural representations of the task-relevant sensory information, or of the upcoming participant's response. We then used these representations in a neuro-behavioral analysis to probe which neural representations of acoustic or visual spatial information are directly predictive of the participant-specific VE and VAE single trial biases.

We applied linear discriminant analysis to the time-resolved MEG source data to determine neural representations of the spatial lateralization of the auditory and visual stimuli (*Figure 2*). From the MEG activity during the A trials, we obtained significant classification (cluster-based permutation test, correcting for multiple comparisons, for details refer to Materials and methods - Statistical Analysis) performance for the current sound ($A_A$; peaking at 80 ms in the left inferior parietal and at 160 ms in the middle temporal gyrus) and for the location of the sound in the previous trial ($A_{AV}$; peaking around 120 ms in the left middle occipital lobe and the bilateral precuneus; at $p \leq 0.01$ FWE corrected for multiple comparisons in source space). This characterizes neural representations of acoustic spatial information currently received and persisting from the previous trial in a wider network of temporal and parietal brain regions. Classification of the lateralization of previous visual stimuli ($V_{AV}$) was not significant at the whole brain level in the activity of the A trial, suggesting that persistent visual information was weaker than that of the acoustic information. However, the whole brain classification maps revealed meaningful clusters in early left inferior temporal areas and the right inferior/superior parietal areas. Classification of the upcoming response ($R_A$) was significant with a similar pattern as observed for the current sound ($A_A$) in the A trial.

## Neural correlates of the VAE

To reveal the neural correlates of the VAE we investigated three regression models capturing different aspects of how current and previous sensory information shape i) the neural encoding of current sensory information in the A trial (i.e. $A_A$), ii) the encoding of the upcoming response ($R_A$), and iii) how neural representations of previous sensory information contribute to the single trial VAE bias.

The first model tested how the previous stimuli affect the encoding of the current sound, that is, how the encoding of sound $A_A$ in the MEG activity of the A trial was affected by the previous stimulus positions $A_{AV}$ and $V_{AV}$ (*Figure 3A*; *Table 1*). There was a significant (cluster-based permutation test FWE corrected at $p \leq 0.05$) influence of $A_{AV}$, starting around 80 ms in the cingulum, precuneus, shifting towards inferior/superior parietal areas around 220 ms. There was also significant influence of $V_{AV}$ in the left occipital/parietal areas around 160 ms. Importantly, the significant effects from the previous acoustic and visual stimuli overlapped in the left parietal areas (*Figure 3A*; red inset). The second model revealed that the previous stimuli also influenced neural activity discriminative of the participants' response ($R_A$; *Figure 3B*; *Table 1*). In particular, both previous sound and visual

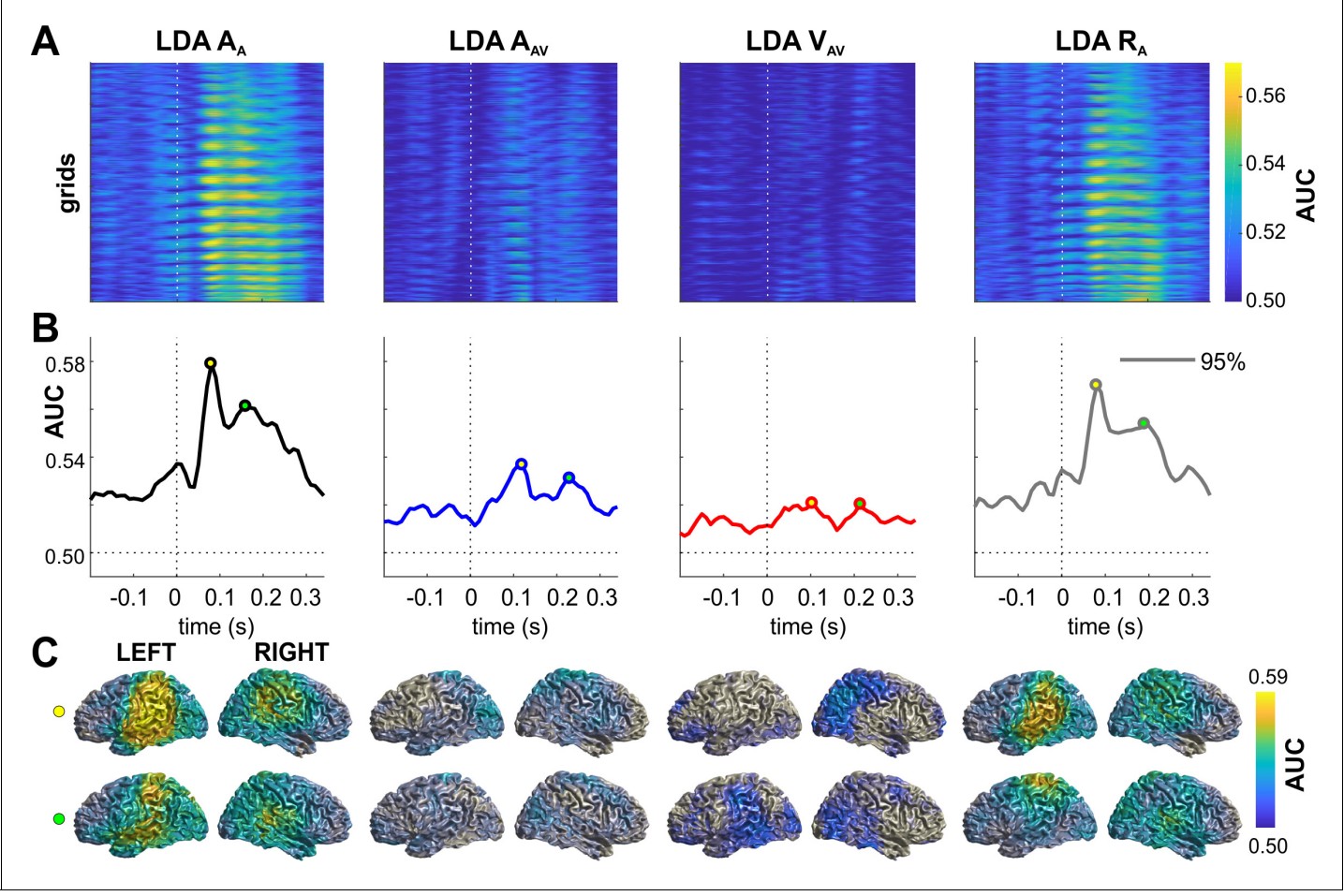

**Figure 2.** Neural representation of current and previous sensory information and upcoming responses. The figure shows the performance (AUC) of linear discriminants for different variables of interest. (**A**) Time-course of discriminant performance for all grid points in source space. (**B**) Time-course of the 95th percentile across source locations. (**C**) Surface projections of significant ($p \leq 0.01$; FWE corrected across multiple tests using cluster-based permutation) performance at the peak times extracted from panel B (open circles). The performance of LDA $V_{AV}$ was not significant when tested across all source locations, and the maps for $V_{AV}$ are not masked with significance. $A_A$: sound location in A trial. $A_{AV}$: sound location in AV trial. $V_{AV}$: visual location in AV trial. Deposited data: Atrial_LDA_AUC.mat.

DOI: https://doi.org/10.7554/eLife.47001.004

stimulus influenced the activity predictive of the current response around 80 ms in the right parietal cortex (precuneus in particular), with the effect of $A_{AV}$ including also frontal and temporal regions. The significant effects of $A_{AV}$ and $V_{AV}$ overlapped in the cingulum and precuneus (*Figure 3B*; red inset). These results demonstrate that parietal regions represent information about previous multi-sensory stimuli, and this information affects the neural encoding of the currently perceived sound.

Using the third model, we directly tested whether these neural signatures of the previous stimuli in the MEG activity during the A trial are significantly related to the participants' single trial response bias (*Figure 3C*; *Table 1*). The significant influences of the neural representations of previous acoustic and visual stimuli overlapped again in parietal cortex (angular gyrus, precuneus; *Figure 3C*; red inset). The converging evidence from these three analyses demonstrates that the same parietal regions retain information about both previously received acoustic and visual spatial information, and that single trial variations in these neural representations directly influence the participants' bias of subsequent sound localization.

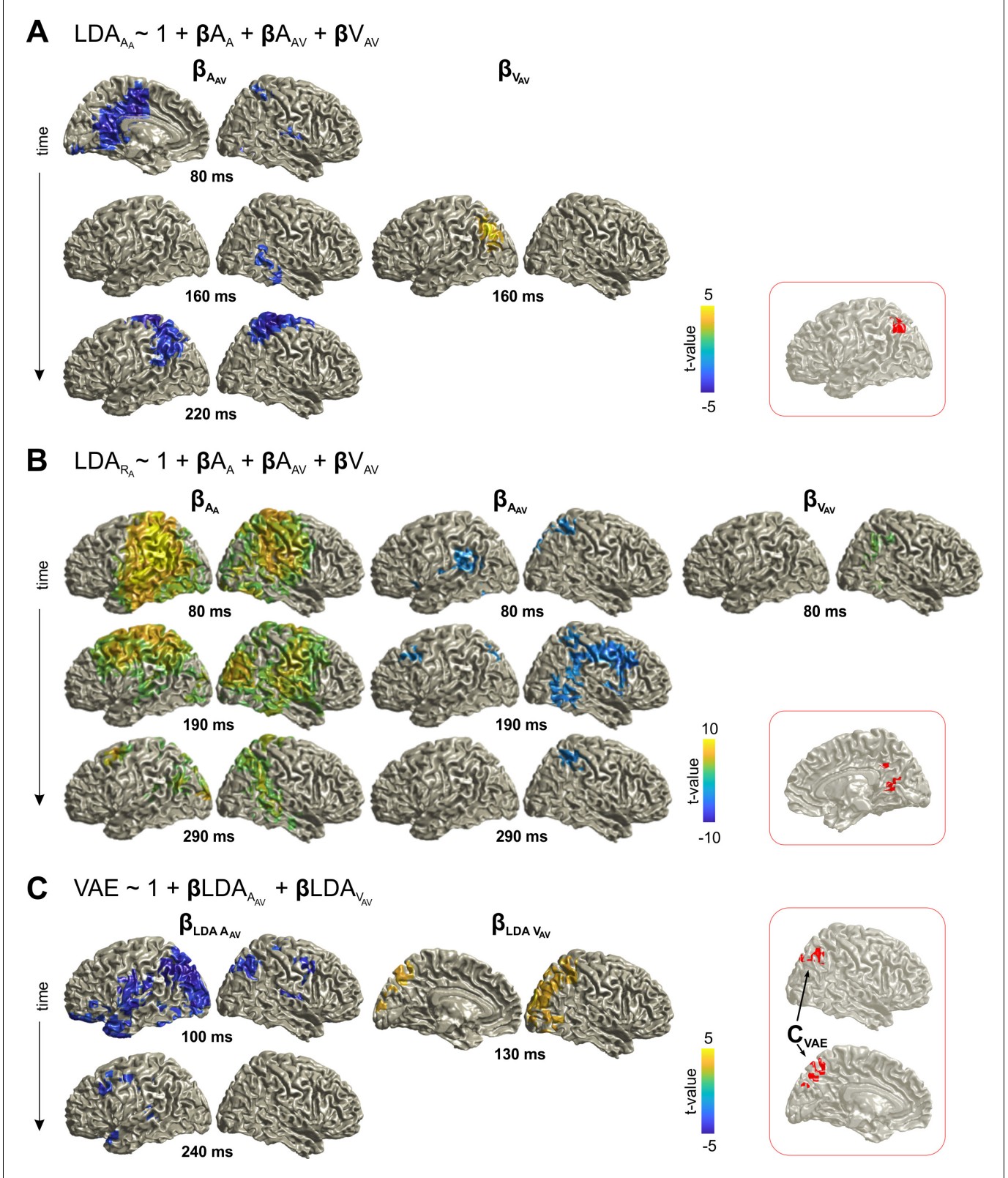

**Figure 3.** Neural correlates of trial-by-trial recalibration (VAE bias). (**A**) Contribution of previous stimuli to the neural representation of the sound ($A_A$) in the A trial (here the effect for $A_A$ itself is not shown). (**B**) Contribution of current and previous stimuli to the neural representation of the response ($R_A$) in the A trial. (**C**) Ventriloquist–aftereffect in the A trial predicted by the neural representation of information about previous stimuli. Red insets: Grid points with overlapping significant effects for both $A_{AV}$ and $V_{AV}$ (**A**, **B**), and for both $LDA_{A\_AV}$ and $LDA_{V\_AV}$ (**C**) across time. Surface projections were

*Figure 3 continued on next page*

*Figure 3 continued*

obtained from whole-brain statistical maps (at p≤0.05, FWE corrected). See *Table 1* for detailed coordinates and statistical results. $A_A$: sound location in A trial. $A_{AV}$: sound location in AV trial. $V_{AV}$: visual location in AV trial. Deposited data: Atrial_LDA_AUC.mat; Atrial_LDA_beta.mat; VAE_beta.mat.
DOI: https://doi.org/10.7554/eLife.47001.005

## Neural correlates of the VE

To be able to directly compare the neural correlates of the ventriloquist aftereffect to multisensory integration (i.e. the VE effect), we repeated the same analysis focusing on the MEG activity in the AV trial. As expected from the above, classification for both auditory ($A_{AV}$) and visual ($V_{AV}$) locations was significant in a network of temporal and occipital regions (*Figure 4—figure supplement 1*). To directly link the encoding of multisensory information to behavior, we again modeled the single trial VE response bias based on the representations of current acoustic and visual information (*Figure 4*). This revealed overlapping representations of both stimuli that directly correlated with the response bias within superior parietal regions (precuneus and superior parietal lobule), and, in a separate cluster, within inferior temporal areas (*Figure 4*; *Table 2*).

## The same parietal regions contribute to integration within a trial and recalibration between trials

The above reveals neural representations of audio-visual information in parietal regions that either contribute to integration within a trial (VE bias) or that shape the localization of auditory information based on previous sensory information (VAE bias). Given that each effect was localized independently (as overlapping clusters in *Figure 3C* and *Figure 4*, respectively), we asked whether the same neural sources significantly contribute to both effects. To this end we subjected the above identified clusters to both neuro-behavioral models (VAE and VE; *Equations 4/5*) to assess the significance of each regressor and to compare the strength of the VAE and VE effects between clusters (*Table 3*).

This revealed that the spatially selective activity contributing to the VE effect ($C_{PAR}$, from *Figure 4*) also significantly contributes to the VAE effect. That is, the single trial variations in the encoding of auditory and visual information in this cluster also contributed significantly (at p≤0.05) to the recalibration effect. Further, the effect strength in this cluster for recalibration did not differ from that observed in the cluster directly identified as significantly contributing to the VAE bias ($C_{VAE}$, at p<0.05; FDR adjusted; *Table 3*). Vice versa, we found that the parietal sources mediating recalibration (cluster $C_{VAE}$; from *Figure 3C*) also significantly contributed to sensory integration within the AV trial (*Table 3*). These results confirm that spatially selective activity within superior parietal regions (identified by both clusters, $C_{PAR}$, and $C_{VAE}$, comprising precuneus and superior parietal regions) is significantly contributing to both sensory integration and trial-by-trial recalibration.

## Hemispheric lateralization of audio-visual integration

While the cluster predictive of the recalibration effect comprised significant grid points in both hemispheres, the model predicting the VE bias in the AV trial based on brain activity was significant only within the left hemisphere (clusters $C_{TEMP}$ and $C_{PAR}$). We performed an additional analysis to directly test whether this effect is indeed lateralized in a statistical sense, that is whether the underlying effect is significantly greater in the left vs. the right hemisphere. First, we compared the ability to discriminate stimulus locations (AUC values) between the actual cluster and the corresponding grid points in the opposite hemisphere: there was no significant difference for either cluster for discriminating the auditory ($A_{AV}$)($C_{TEMP}$; p=0.10, $C_{PAR}$; p=0.65, FDR corrected) or visual stimulus locations ($V_{AV}$)($C_{TEMP}$; p=0.24, $C_{PAR}$; p=0.35, FDR corrected). Second, we compared the contributions of each cluster to the VE bias. The auditory contribution (regression beta for $A_{AV}$) differed significantly between hemispheres for $C_{TEMP}$ (p=0.03, FDR corrected) but not for $C_{PAR}$ (p=0.56, FDR corrected). The visual contribution differed for neither cluster (regression beta for $V_{AV}$, p=0.68, FDR corrected). Hence, the overall evidence for the neural correlates of the VE bias to be lateralized was weak, and absent for the parietal contribution.

**Table 1.** Neuro-behavioral modeling of the VAE.

The significance of each predictor was tested at selected time points at the whole-brain level ($p \leq 0.05$, FWE corrected). The table provides the peak coordinates of significant clusters, the anatomical regions contributing to significant clusters (based on the AAL Atlas), as well as beta and cluster-based t-values (df = 23). The overlap was defined as grid points contributing to both a significant effect for $A_{AV}$ and $V_{AV}$ (at any time). The effect of $A_A$ is not indicated, as this was significant for a large part of the temporal and parietal lobe, and was not of primary interest. L: left hemisphere; R: right hemisphere. BA: Brodmann area. **sum of 2 spatially separate clusters, ***sum of 4 clusters.

| Regressor | Post-stim. time (ms) | Anatomical labels | MNI coord. (peak) Brodmann Area | β t-value ($t_{sum}$) |
|---|---|---|---|---|
| | | $LDA_{A\_A} \sim 1 + A_A + A_{AV} + V_{AV}$ | | |
| $A_{AV}$ | 80 | L/R: Cingulum Mid., Precuneus<br>L: Supp. Motor Area | −3,−20, 29<br>BA 23 | −5.6<br>(−1220) |
| | 160 | R: Fusiform, Temporal Mid/Inf | 28,−39, −20<br>BA 37 | −3.8<br>(−203) |
| | 220 | L/R: Postcentral<br>L: Parietal Inf/Sup<br>R: Precentral, Supp. Motor Area | −24,−36, 77<br>BA 03 | −6.6<br>(−1560) |
| $V_{AV}$ | 160 | L: Occipital Mid/Sup, Parietal Inf/Sup | −40,−76, 37<br>BA 19 | 5.3<br>(246) |
| overlap | - | L: Angular, Parietal Inf., Occipital Mid. | −40,−62, 47<br>BA 39 | - |
| | | $LDA_{R\_A} \sim 1 + A_A + A_{AV} + V_{AV}$ | | |
| $A_A$ | 80 | L: Angular, Temporal Mid.<br>L/R: Postcentral, Precuneus | −40,−52, 21<br>BA 39 | 17.9<br>(13862) |
| | 190 | R: Precuneus, Lingual<br>Temporal Sup.<br>L: Pre/Postcentral, Precuneus | 8,−51, 5<br>BA 30 | 9.3<br>(8323) |
| | 290 | L: Occipital Mid/Sup.<br>R: Temporal Mid/Sup., Parietal Sup. | −24,−100, 5<br>BA 17 | 6.9<br>(2390) |
| $A_{AV}$ | 80 | L: Lingual, Precuneus<br>R: Lingual, Parietal Sup. | −24,−52, 13<br>BA 17 | −5.8<br>(−1152) |
| | 190 | R: Frontal Mid/Inf, Precentral | 33, 21, 21<br>BA 48 | −6.1<br>(−1343) |
| | 290 | R: Precuneus, Cingulum Mid/Post.<br>Parietal Inf/Sup., Postcentral | 16,−42, 41<br>BA 23 | −4.2<br>(−256) |
| $V_{AV}$ | 80 | R: Fusiform, Angular, Parietal Inf. Temporal Mid.Inf, Precuneus | 32,−51, −3<br>BA 37 | 4.3<br>(225) |
| overlap | - | R: Calcarine, Precuneus, Cingulum Mid. | 17,−67, 23<br>BA 18 | - |
| | | $VAE \sim 1 + LDA_{A\_AV} + LDA_{V\_AV}$ | | |
| $LDA_{A\_AV}$ | 100 | L: Occipital Mid/Sup., Temporal Inf., Parietal Mid/Sup<br>L/R: Precuneus | −32,−87, 37<br>BA 19 | −7.4<br>(−3571)** |
| | 240 | L: Precentral, Frontal Mid, Precuneus, Temporal Pole Sup | −32,−20, 45<br>BA 03 | −4.7<br>(−413)*** |
| $LDA_{V\_AV}$ | 130 | R: Occipital Mid/Sup, Parietal Inf/Sup, Angular | 24,−95, 29<br>BA 18 | 4.7<br>(579) |
| overlap ($C_{VAE}$) | - | L/R: Precuneus<br>R: Angular | −3,−65, 51<br>BA 07 | - |

DOI: https://doi.org/10.7554/eLife.47001.006

## Parietal and temporal regions encode combined multisensory information

The results so far demonstrate that medial superior parietal activity reflects both, integration and recalibration. Given that the behavioral recalibration in the A trial was driven by the combined audio-visual information in the preceding AV trial (c.f. *Figure 1F*) this raises the question as to

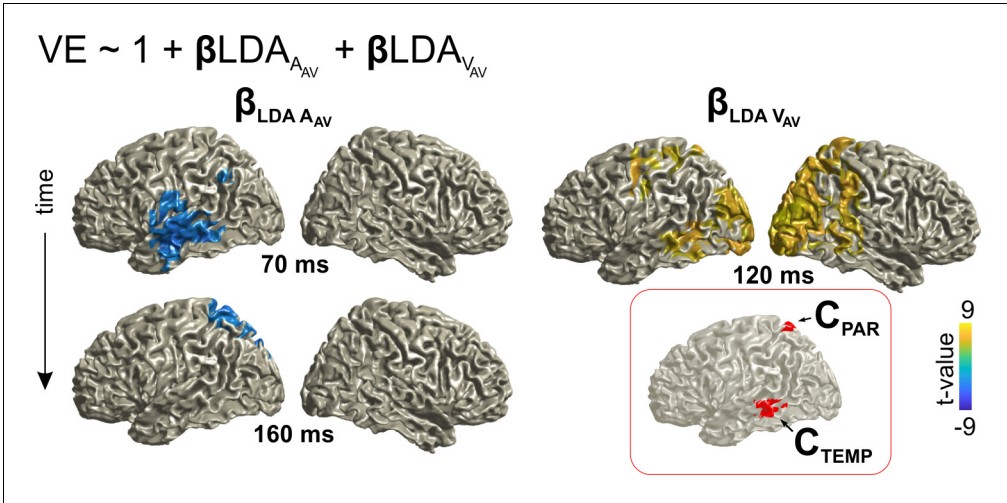

**Figure 4.** Neural correlates of audio-visual integration within a trial (VE bias). Contribution of the representations of acoustic and visual information to the single trial bias in the AV trial. Red inset: Grid points with overlapping significant effects for both LDA$_{A\_AV}$ and LDA$_{V\_AV}$. Surface projections were obtained from whole-brain statistical maps (at p≤0.05, FWE corrected). See *Table 2* for detailed coordinates and statistical results. A$_{AV}$: sound location in AV trial. V$_{AV}$: visual location in AV trial. Deposited data: AVtrial_LDA_AUC.mat; VE_beta.mat.
DOI: https://doi.org/10.7554/eLife.47001.007
The following figure supplement is available for figure 4:

**Figure supplement 1.** Neural representation of sensory information in AV trials.
DOI: https://doi.org/10.7554/eLife.47001.008

whether the neurally encoded information (in the MEG activity in the A trial) about the previous stimuli (from the AV trial) reflects previous unisensory information, or the behaviorally combined information. To test this, we compared the classification performance of the MEG activity in each cluster of interest (C$_{VAE}$, C$_{TEMP}$, C$_{PAR}$) for the location of the previous sound (A$_{AV}$) and the combined sensory

**Table 2.** Neuro-behavioral modeling of the VE.
The significance of each predictor was tested at selected time points at the whole-brain level (p≤0.05, FWE corrected). The table provides the peak coordinates of significant clusters, the anatomical regions contributed to significant clusters (based on the AAL Atlas), peak beta values and cluster-based t-values (df = 23). The overlap was defined as grid points contributing to both a significant effect for LDA$_{A\_AV}$ and LDA$_{V\_AV}$ (at any time). L: left hemisphere; R: right hemisphere. BA: Brodmann area. **sum of 2 spatially separate clusters.

| | VE ~ 1 + LDA$_{A\_AV}$ + LDA$_{V\_AV}$ | | | |
|---|---|---|---|---|
| **Regressor** | **Post-stim. time (ms)** | **Anatomical labels** | **MNI coord. (peak) Brodmann Area** | **β t-value (t$_{sum}$)** |
| LDA$_{A\_AV}$ | 70 | L: Temporal Mid/Sup., Rolandic Oper, Postcentral, Heschl | −47,−19, −19 BA 20 | −4.1 (−392) |
| | 160 | L: Parietal Inf/Sup., Precuneus, Cuneus Occipital Sup | −24,−60, 69 BA 07 | −4.0 (−181) |
| LDA$_{V\_AV}$ | 120 | L/R: Occipital Mid., Calcarine R: Occipital Sup., Temporal Mid., Lingual, Cuneus | 24,−92, 13 BA 18 | 9.1 (7188)** |
| overlap (C$_{TEMP}$, C$_{PAR}$) | - | L: Temporal Mid Parietal Sup, Cuneus, Precuneus | −58,−41, −6 (C$_{TEMP}$, BA 21) −14,−60, 70 (C$_{PAR}$, BA 05, 07) | - |

DOI: https://doi.org/10.7554/eLife.47001.009

**Table 3.** Overlapping neural substrates for integration and recalibration.

Both neuro-behavioral models, VE and VAE (*Equations 4/5*), were tested within the clusters significantly contributing to the VAE effect (from *Figure 3C*, $C_{VAE}$) and the two clusters contributing to the VE effect (from *Figure 4*, $C_{TEMP}$, $C_{PAR}$). The table lists regression betas and group-level t-values. The expected effects (based on *Figure 3C* and *Figure 4*) are shown in normal font, the effects of interest (cross-tested) in BOLD. We directly compared the effect strengths between clusters (one-sided paired t-test, p<0.05, FDR adjusted). Significant results are indicated by *. In particular, both $C_{VAE}$ and $C_{PAR}$ have significant VAE and VE effects (tcrit = 2.81, and their respective effect sizes do not differ between clusters (ns beta differences).

| Model | VAE ~ $1 + \beta*LDA_{A\_AV} + \beta*LDA_{V\_AV}$ | | | VE ~ $1 + \beta*LDA_{A\_AV} + \beta*LDA_{V\_AV}$ | |
|---|---|---|---|---|---|
| Cluster | $C_{VAE}$ | $C_{TEMP}$ | $C_{PAR}$ | $C_{VAE}$ | $C_{PAR}$ |
| t-value ($\beta_{LDAA\_AV}$) | −3.29 (−0.12) | −2.86 (−0.13)ns | −2.97 (−0.09)ns | **−2.50 (−0.30)ns** | −3.31 (−0.32) |
| t-value ($\beta_{LDAV\_AV}$) | 3.17 (0.12) | 0.91 (0.03)* | 2.23 (0.08)ns | **6.61 (3.88)ns** | 6.93 (3.35) |

DOI: https://doi.org/10.7554/eLife.47001.010

information, as predicted by each participant's behavioral weighting model (i.e., the VE bias predicted by $mi_3$: $\beta_s*A_{AV} + \beta_s*V_{AV}$, s: participant c.f. *Figure 1B*).

For parietal activity in the AV trial (clusters $C_{VAE}$ and $C_{PAR}$) discriminant performance was significantly higher for the weighted multisensory than for unisensory $A_{AV}$ information (two-sided paired t-test, $p \leq 3*10^{-6}$, for both comparisons, FDR corrected; *Figure 5*), confirming that these regions indeed encode the integrated multisensory information. This difference was no longer significant when tested using the brain activity in the A trial, possibly because the overall classification performance was lower for previous than for current stimuli. In contrast, temporal activity ($C_{TEMP}$) was

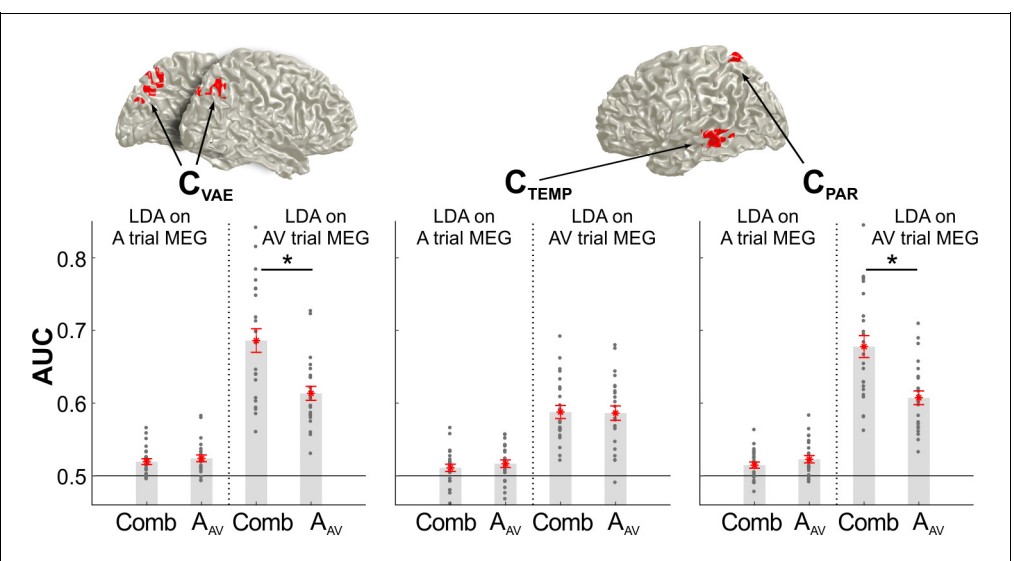

**Figure 5.** Classification performance for unisensory and combined multisensory information. The bar graphs show the classification performance for each cluster of interest ($C_{VAE}$ from *Figure 3C*, $C_{PAR}$ and $C_{TEMP}$ from *Figure 4*) based on the activity in the AV trial or the A trial. Classification was applied to either the sound location in the AV trial ($A_{AV}$), or the combined multisensory information in the AV trial (Comb), derived from the participant specific VE bias (derived from model $mi_3$,VE ~ $1 + \beta \cdot A_{AV} + \beta \cdot V_{AV}$ for the behavioral data). Asterisks denote p<0.01, two-sided paired t-test, FDR corrected for multiple comparisons at $p \leq 0.05$. Gray dots are individual participant values averaged within each cluster, red stars are the mean across participants, and red lines are standard errors of mean. $A_{AV}$: sound location in AV trial. $V_{AV}$: visual location in AV trial. Deposited data: LDA_AUC_comb.mat.
DOI: https://doi.org/10.7554/eLife.47001.011

equally sensitive to unisensory and combined multisensory information in both trials, in line with temporal regions participating both in sensory integration and unisensory processing (*Beauchamp et al., 2004*).

### Eye movements and the encoding of spatial information

Given potential influences of eye position on behavioral sound localization (*Kopco et al., 2009*; *Razavi et al., 2007*) it is important to rule out that eye movements consistently influenced the above results. Because eye tracking was technically impossible due to the close participant-to-screen distance, which was essential in order to promote audio-visual co-localization, we used the MEG data to confirm that participants indeed maintained fixation. We extracted ICA components known to reflect eye movement related artifacts based on their topographies and time courses (*Hipp and Siegel, 2013*; *Keren et al., 2010*) and determined the presence of potential EOG signals during the fixation-, stimulus-, and post-stimulus periods. This revealed that only 4.9 ± 1.7% (mean ± s.e.m) of trials exhibited evidence of potential eye-movements across all participants. In particular, eye-movements during the stimulus presentation period were rare (0.35 ± 0.12%, max 2.2%). Given that participants fixated the central fixation dot in both AV and A trials, this suggests that sound localization performance, or the VAE bias, are not affected by systematic differences in eye position across trials.

## Discussion

Adaptive behavior in multisensory environments requires the combination of relevant information received at each moment in time, and the adaptation to contextual changes over time, such as discrepancies in multisensory evidence. This multi-facetted profile of flexible multisensory behavior raises the question of whether the sensory integration at a given moment, and the use of previous multisensory information to recalibrate subsequent perception, arise from the same or distinct neural mechanisms. We here directly compared multisensory integration and trial-by-trial recalibration in an audio-visual spatial localization paradigm. Using single trial MEG analysis we determined a network of temporal and parietal brain regions that mediate behavioral sound localization. Of these regions, superior medial parietal activity represents current auditory and visual information, encodes the combined multisensory estimate as reflected in participant's behavior, and retains information during the subsequent trial. Importantly, these parietal representations mediate both the multisensory integration within a trial and the subsequent recalibration of unisensory auditory perception, suggesting a common neural substrate for sensory integration and trial-by-trial recalibration of subsequent unisensory perception.

### Neural signatures of previous sensory information

Despite many behavioral studies demonstrating the robustness of multisensory trial-by-trial recalibration (*Bosen et al., 2017*; *Bruns and Röder, 2015*; *Wozny and Shams, 2011a*), little is known about the underlying neural substrate. Reasoning that recalibration relies on the persistence of information about previous stimuli, our study was guided by the quantification of where in the brain sensory information experienced during one trial can be recovered during the subsequent trial. This revealed persistent representations of previous acoustic information in a temporal-parietal network, suggesting that the regions known to reflect auditory spatial information within a trial also retain previous sensory evidence to mediate behavior (*At et al., 2011*; *Bizley and Cohen, 2013*; *Lewald et al., 2008*; *Zimmer and Macaluso, 2005*).

Neural representations of previous visual information were also strongest in temporal and parietal regions, although classification performance was not significant at the whole brain level. Importantly, the behavioral data clearly revealed a lasting effect of visual information on behavior. Furthermore, the neuro-behavioral analysis revealed a significant contribution to behavior of neural representations of previous visual information in the superior parietal cortex. One reason for the weaker classification performance for previous visual stimuli could be a bias towards acoustic information in the participant's task, which was to localize the sound rather than the visual stimulus in both trials. Alternatively, it could be that the visual information is largely carried by neural activity reflecting the combined audio-visual information, and hence persists directly in form of a genuine multisensory representation. This integrated multisensory representation comprises a stronger component of

acoustic over visual spatial information, as reflected by the ventriloquist bias in the present paradigm. Our data indeed support this conclusion, as parietal activity within the audio-visual trial was encoding the behaviorally combined information more than the acoustic information. Previous work has shown that parietal activity combines multisensory information flexibly depending on task-relevance and crossmodal disparity (*Eramudugolla et al., 2011*; *Rohe and Noppeney, 2016*; *Rohe and Noppeney, 2018*) and the focus on reporting the sound location in our task may have led to the attenuation of the combined visual information in the subsequent trial. An interesting approach to test this could be to reverse the VE and thus task-relevance with very blurred visual stimulus (*Alais and Burr, 2004*) and investigate if a symmetry in the VAE and its neural correlates hold. Furthermore, it is also possible, that eye movements during the response period may have contributed to reducing a persistent representation of visual information, in part as additional visual information was seen and processed during the response period (e.g. the response cursor). Future work is required to better understand the influence of task-relevance on uni- and multisensory representations and how these are maintained over time.

## Multiple facets of multisensory integration in parietal cortex

Several brain regions have been implied in the merging of simultaneous audio-visual spatial information (*Atilgan et al., 2018*; *Bizley et al., 2016*; *Sereno and Huang, 2014*). Our results suggest that the perceptual bias induced by vision on sound localization (i.e. the ventriloquist effect) is mediated by the posterior middle temporal gyrus and the superior parietal cortex, with parietal regions encoding the perceptually combined multisensory information. These regions are in line with previous studies, which have pinpointed superior temporal regions, the insula and parietal-occipital areas as hubs for multisensory integration (*Bischoff et al., 2007*; *Bonath et al., 2014*; *Callan et al., 2015*; *Bonath et al., 2007*). In particular, a series of studies revealed that both temporal and parietal regions combine audio-visual information in a reliability- and task-dependent manner (*Aller and Noppeney, 2019*; *Rohe et al., 2019*; *Rohe and Noppeney, 2015b*; *Rohe and Noppeney, 2016*). However, while posterior parietal regions reflect the automatic fusion of multisensory information, more anterior parietal regions reflect an adaptive multisensory representation that follows predictions of Bayesian inference models. These anterior regions (sub-divisions IPS3 and 4 in *Rohe and Noppeney, 2015b*) combine multisensory information when two cues seem to arise from a common origin and only partially integrate when there is a chance that the two cues arise from distinct sources, in accordance with the flexible use of discrepant multisensory information for behavior (*Cao et al., 2019*; *Körding et al., 2007*). Noteworthy, the peak effect for the ventriloquist aftereffect found here was located at the anterior-posterior location corresponding to the border of IPS2 and IPS3 (*Wang et al., 2015*), albeit more medial. While the significant clusters were more pronounced on the left hemisphere, a direct assessment did not provide evidence for these effects to be lateralized in a strict sense (*Liégeois et al., 2002*). Our results hence corroborate the behavioral relevance of superior-anterior parietal representations and fit with an interpretation that these regions mediate the flexible use of multisensory information, depending on task and sensory congruency, to mediate adaptive behavior.

In contrast, very little is known about the brain regions implementing the trial-by-trial recalibration of unisensory perception by previous multisensory information. In fact, most studies have relied on prolonged adaptation to multisensory discrepancies. Hence these studies investigated long-term recalibration, which seems to be mechanistically distinct from the trial-by-trial recalibration investigated here (*Bosen et al., 2017*; *Bruns et al., 2011*; *Bruns and Röder, 2010*; *Bonath et al., 2007*; *Zierul et al., 2017*). The study most closely resembling the present one suggested that the ventriloquist after-effect is mediated by an interaction of auditory and parietal regions (*Zierul et al., 2017*), a network centered view also supported by work on the McGurk after-effect (*Kilian-Hütten et al., 2011a*; *Kilian-Hütten et al., 2011b*). Yet, no study to date has investigated the direct underpinnings of multisensory recalibration at the trial-by-trial level, or attempted to directly link the neural signature of the encoded sensory information about previous stimuli to the participant-specific perceptual bias. Our results close this gap by demonstrating the behavioral relevance of anterior medial parietal representations of previous multisensory information, which seem to reflect the flexible combination of spatial information following multisensory causal inference, and have a direct influence on participants' perceptual bias in localizing a subsequent unisensory stimulus.

The retention of information about previous stimuli in parietal cortex directly links to animal work, which has revealed a mixed pattern of neural selectivity in parietal cortex, with individual neurons encoding both unisensory and multisensory information, reflecting the accumulation of this over time, and the transformation into perceptual choice (*Raposo et al., 2014*). For example, parietal neurons are involved in maintaining the history of prior stimulus information, a role that is directly in line with our results (*Akrami et al., 2018*). Also, the observation that the previously experienced stimuli shape both the neural encoding of the subsequent sound and neural correlates of the upcoming response (*Figure 3A,B*) suggests that these representations of prior evidence emerge in a neural system intermediate between pure sensory and pure motor (or choice) representations. Again, this fits with the observation of mixed neural representation in parietal neurons. Our results set the stage to directly probe the correlates of multisensory recalibration at the single neuron level, for example, to address whether integration and recalibration are mediated by the very same neural populations.

In humans, the medial superior parietal cortex pinpointed here as mediator of recalibration has been implied in maintaining spatial and episodic memory (*Müller et al., 2018*; *Pollmann et al., 2003*; *Schott et al., 2018*; *Uncapher and Wagner, 2009*; *Vilberg and Rugg, 2008*) used for example during navigation, spatial updating or spatial search (*Brodt et al., 2016*; *Pollmann et al., 2003*). Our results broaden the functional scope of these parietal regions in multisensory perception, by showing that these regions are also involved in the integration of multiple simultaneous cues to guide subsequent spatial behavior. This places the medial parietal cortex at the interface of momentary sensory inference and memory, and exposes multisensory recalibration as a form of implicit episodic memory, mediating the integration of past and current information into a more holistic percept.

## Integrating multisensory information across multiple time scales

Similar to other forms of memory, the history of multisensory spatial information influences perception across a range of time scales (*Bosen et al., 2017*; *Bosen et al., 2018*; *Bruns and Röder, 2015*). In particular, recalibration emerges on a trial-by-trial basis, as investigated here, and after several minutes of exposure to consistent discrepancies (*Frissen et al., 2012*; *Kopco et al., 2009*; *Mendonça et al., 2015*; *Radeau and Bertelson, 1974*; *Razavi et al., 2007*; *Recanzone, 1998*). Behavioral studies have suggested that the mechanisms underlying the trial-by-trial and long-term effects may be distinct (*Bruns and Röder, 2015*; *Bruns and Röder, 2017*). Yet, it remains possible that both are mediated by the same neural mechanisms, such as the same source of spatial memory (*Müller et al., 2018*; *Schott et al., 2018*). Indeed, an EEG study on multisensory long-term recalibration has reported neural correlates compatible with an origin in parietal cortex (*Bruns et al., 2011*). The finding that medial parietal regions are involved in spatial and episodic memory and mediate trial-by-trial perceptual recalibration clearly lends itself to hypothesize that the very same regions should also contribute to long term recalibration as well.

## The role of coordinate systems in spatial perception

Eye movement patterns can affect sound localization. For example, changes in fixation affect early auditory cortex (*Fu et al., 2004*; *Werner-Reiss et al., 2003*), visual prism-adaptation results in changes in sound localization behavior (*Zwiers et al., 2003*), and both prolonged peripheral fixation and the continuous use of fixation while searching for sounds can bias sound localization (*Razavi et al., 2007*). This raises the question as to whether audio-visual recalibration emerges in head- or eye-centered coordinate systems (*Kopco et al., 2009*). Spatial representations in the brain emerge in a number of coordinate systems (*Chen et al., 2018*; *Schechtman et al., 2012*; *Town et al., 2017*) raising the possibility that visual and auditory information is combined in any, or possibly multiple, coordinate systems. Indeed, one study suggested that the long-term ventriloquist after-effect arises in mixed eye-head coordinates (*Kopco et al., 2009*). However, this study also noted that a transition from head- to mixed-spatial representations emerges slowly and only after prolonged exposure, while recalibration after a few exposure trials is best explained in head coordinates. With respect to the trial-by-trial recalibration investigated here, this suggests an origin in head-centered coordinates. The present study was not designed to disentangle different types of spatial representations, as the participants maintained fixation before, during and after the stimulus, and hence eye and head coordinates were aligned. Any, if present, systematic biases in fixations

were eliminated on a trial-by-trial level by randomizing stimulus locations across trials and allowing participants to move their eyes while responding (*Razavi et al., 2007*). This makes it unlikely that systematic patterns of eye movements would have affected our results, and calls for further work to elucidate the precise coordinate systems encoding the different forms or recalibration.

The use of virtual sound locations, rather than for example an array of speakers, may have affected the participants' tendency to bind auditory and visual cues (*Fujisaki et al., 2004*). While the use of HRTFs is routine in neuroimaging studies on spatial localization (*Rohe and Noppeney, 2015b*), individual participants may perceive sounds more 'within' the head in contrast to these being properly externalized. While this can be a concern when determining whether audio-visual integration follows a specific (e.g. Bayes optimal) model (*Meijer et al., 2019*), it would not affect our results, as these are concerned with relating the trial specific bias expressed in participants behavior with the underlying neural representations. Even if visual and acoustic stimuli were not perceived as fully co-localized, this may have reduced the overall ventriloquist bias, but would not affect the neuro-behavioral correlation. Indeed, the presence of both the ventriloquist bias and the trial-by-trial recalibration effect suggests that participants were able to perceive the spatially disparate sound sources, and co-localize the sound and visual stimulus when the disparity was small.

## Conclusion

Navigating an ever-changing world, the flexible use of past and current sensory information lies at the heart of adaptive behavior. Our results show that multiple regions are involved in the momentary integration of spatial information, and specifically expose the medial superior parietal cortex as a hub that maintains multiple sensory representations to flexibly interface the past with the environment to guide adaptive behavior.

# Materials and methods

**Key resources table**

| Reagent type (species) or resource | Designation | Source or reference | Identifiers | Additional information |
|---|---|---|---|---|
| Species (Human) | Participants | Volunteers recruited from adverts | | |
| Software, algorithm | MATLAB R2017A | MathWorks | https://www.mathworks.com/ | |
| Software, algorithm | Psychtoolbox-3 | Brainard, 1997; Pelli, 1997 | http://psychtoolbox.org/ | |
| Software, algorithm | SPM8 | Wellcome Trust | http://www.fil.ion.ucl.ac.uk/spm/software/spm8/ | |
| Software, algorithm | FreeSurfer | *Fischl, 2012* | https://surfer.nmr.mgh.harvard.edu/ | |
| Software, algorithm | PKU and IOA HRTF database | *Qu et al., 2009* | http://www.cis.pku.edu.cn/auditory/Staff/Dr.Qu.files/Qu-HRTF-Database.html | |
| Software, algorithm | Fieldtrip | *Oostenveld et al., 2011* | http://www.fieldtriptoolbox.org/ | |
| Other | Behavioral data | This paper | https://dx.doi.org/10.5061/dryad.t0p9c93 | data generated in this study |
| Other | MEG data | This paper | https://dx.doi.org/10.5061/dryad.t0p9c93 | data generated in this study |

Twenty-six healthy right-handed adults participated in this study (15 females, age 24.2 ± 4.7 years). Sample size was determined based on previous studies using similar experimental protocols (*Clarke et al., 2015*; *Clarke et al., 2013*) and recommendation for sample sizes in empirical psychology (*Simmons et al., 2011*). Data from two participants (both females) had to be excluded as these were incomplete due to technical problems during acquisition, hence results are reported for 24

participants. All participants submitted written informed consent, and reported normal vision and hearing, and indicated no history of neurological diseases. The study was conducted in accordance with the Declaration of Helsinki and was approved by the local ethics committee (College of Science and Engineering, University of Glasgow) (Ethics Application No: 300140078).

## Task Design and Stimuli

The paradigm was based on an audio-visual localization task (*Wozny and Shams, 2011a*). Trials and conditions were designed to probe both the ventriloquist effect and the ventriloquist-aftereffect. A typical sequence of trials is depicted in *Figure 1A*. The participants' task was to localize a sound during either Audio-Visual (AV) or Auditory (A), trials, or, on a subset of trials (~8% of total trials), to localize a visual stimulus (V trials). The locations of auditory and visual stimuli were each drawn semi-independently from five locations ($-17°, -8.5°, 0°, 8.5°, 17°$ of visual angle from the midline), to yield nine different audio-visual discrepancies ($-34°, -25.5°, -17°, -8.5°, 0°, 8.5°, 17°, 25.5°, 34°$). Importantly, AV and A trials were always presented sequentially (AV trial preceding A trial) to probe the influence of audio-visual integration on subsequent unisensory perception.

Acoustic stimuli were spatially dispersed white noise bursts, created by applying head related transfer functions (HRTF) (PKU and IOA HRTF database, *Qu et al., 2009*) to white noise (duration = 50 ms) defined at specific azimuths, elevations and distances. Here we used a distance of 50 cm and 0 elevation. The behavioral data obtained during A trials confirm that participants perceived these sounds as lateralized, Sounds were sampled at 48 kHz, delivered binaurally by an Etymotic ER-30 tube-phone at ~84.3 dB (root-mean-square value, measured with a Brüel and Kjær Type 2205 sound-level meter, A-weighted). An inverse filtering procedure was applied to compensate for the acoustic distortion introduced by plastic tubes required for the use in the MEG shield-room (*Giordano et al., 2018*). The visual stimulus was a white Gaussian disk of 50 ms duration covering 1.5° of visual angle (at full-width half-maximum). This was back-projected onto a semi-transparent screen located 50 cm in front of the participant, via a DLP projector (Panasonic D7700). Stimulus presentation was controlled with Psychophysics toolbox (Brainard, 1997) for MATLAB (The MathWorks, Inc, Natick, MA), with ensured temporal synchronization of auditory and visual stimuli.

In total we repeated each discrepancy (within or between trials) 40 times, resulting in a total of 360 AV-A trial pairs. In addition, 70 visual trials were interleaved to maintain attention (V trials always came after A trials, thus not interrupting the AV-A pairs), resulting in a total of 790 trials ($2 \times 360 + 70$) for each participant. Trials were pseudo-randomized, and divided into 10 blocks of ~8 mins each. Each trial started with a fixation period (uniform range 800 ms–1200 ms), followed by the stimulus (50 ms). After a random post-stimulus period (uniform range 600 ms–800 ms) the response cue emerged, and was shown as a horizontal bar along which participants could move a cursor. Participants responded by moving a trackball mouse (Current Designs Inc, Philadelphia, PA 19104 USA) with their right hand by moving the cursor to the location of the perceived stimulus and clicking the button. A letter 'S' was displayed on the cursor for 'sound', and 'V' for the visual trials. There was no constraint on response times. Inter-trial intervals varied randomly (uniform 1100 ms–1500 ms) and the experiment lasted about 3.5 hr including preparation and breaks. Importantly, participants were asked to maintain fixation during the entire pre-stimulus fixation period, the stimulus, and the post-stimulus period until the response cue appeared. During the response itself, they could freely move their eyes.

## Analysis of behavioral data

### Ventriloquist effect (VE)

For AV trials, we defined the VE as the difference between the reported location ($R_{AV}$) and the location at which the sound ($A_{AV}$) was actually presented ($R_{AV} - A_{AV}$). To determine whether the response bias captured by the VE was systematically related to any of the sensory stimuli, we compared the power of different linear mixed models for predicting this responses bias as computational accounts for the VE. These models relied either on only the auditory, only the visual stimulus location, or their combination: $mi_1$: VE ~ 1 + $\beta \cdot A_{AV}$ +subj, $mi_2$: VE ~ 1 + $\beta \cdot V_{AV}$ +subj, $mi_3$: VE ~ 1 + $\beta \cdot A_{AV}$ + $\beta \cdot V_{AV}$ +subj, where $A_{AV}$ and $V_{AV}$ were the main effects, and the participant ID (subj) was included as random effect. Models were fit using maximum-likelihood procedures and we calculated the relative Bayesian information criterion (BIC) (BIC - mean($BIC_m$), and relative Akaike information criterion (AIC)

(AIC - mean($AIC_m$)), and the protected exceedance probability (*Rigoux et al., 2014*) for formal model comparison.

### Ventriloquist-aftereffect (VAE)

The VAE was defined as the difference between the reported location ($R_A$) minus the mean reported location for all trials of the same stimulus position ($R_A$ – mean($R_A$)). This was done to ensure that any overall bias in sound localization (e.g. a tendency to perceive sounds are closer to the midline than they actually are) would not influence this bias measure (*Wozny and Shams, 2011a*). This was then expressed as a function of the audio-visual discrepancy in the previous trial ΔVA (i.e., $V_{AV}$ - $A_{AV}$). Again we used linear mixed-effects models to compare different accounts of how the response bias depends on the stimuli: $mr_1$: VAE ~ 1 + β·$A_{AV}$ +subj, $mr_2$: VAE ~ 1 + β·$V_{AV}$ +subj, $mr_3$: VAE ~ 1 + β·$A_{AV}$ + β·$V_{AV}$ +subj. We also quantified the response bias as a function of the all stimulus locations (i.e. $A_A$, $V_{AV}$, $A_{AV}$), in order to determine the influence of each individual stimulus on behavior (*Figure 1C*).

## Magnetoencephalography (MEG) acquisition

Participants were seated in a magnetically shielded room, 50 cm in front of a screen (30.5 cm x 40.5 cm, 1024 × 768 resolution). The MEG data was recorded with a 248-magnetometer, whole-head MEG system (MAGNES 3600 WH, 4-D Neuroimaging, San Diego, CA) at a sampling rate of 1017.25 Hz. Head positions were measured at the beginning and end of each block, using five coils marking fiducial landmarks on the head of the participants, to monitor head movements. Coil positions were co-digitized with the head shape (FASTRAK, Polhemus Inc, Colchester, VT). Mean head movement across all participants for all blocks was 2.7 mm ± 0.2 mm (mean ± s.e.m.).

## Preprocessing of MEG data

MEG data were preprocessed with MATLAB (The MathWorks, Inc, Natick, MA) using the Fieldtrip toolbox (version 20171001, *Oostenveld et al., 2011*). Each block was preprocessed individually (ft_preprocessing). Epochs of −0.6 s ~ 0.6 s (0 = stimulus onset) were extracted from the continuous data, and denoised using the MEG reference (ft_denoise_pca). Resulting data was filtered between 1 ~ 48 Hz (4-order Butterworth filter, forward and reverse), and down-sampled to 100 Hz. Known faulty channels (N = 3) were removed. Then, variance, maximum, minimum, and range of data across trials were calculated for each channel, and channels with extreme data were excluded. Outliers were defined based on interquartiles (IQR) (*Tukey, 1977*); Q1 – 4.5 × IQR or above Q3 +4.5 × IQR. Weight 4.5 instead of the standard 1.5 was used since 1.5 was eliminating too many channels. Overall about 6% of all channels were excluded (15.1 ± 5.8 channels per participant; mean ± SD). Heart and eye-movement artifacts were removed using independent component analysis (ICA) with Fieldtrip (ft_componentanalysis, ft_rejectcomponent), which was calculated based on 30 principal components. Trials with SQUID (superconducting quantum interference device) jumps were detected and removed (ft_artifact_jump) with a cutoff z-value of 20. Finally, the data was manually inspected using the interquartile method (across channels weight 2.5) to exclude outlier trials. On average about 2% of trials had to be discarded (18.3 ± 7.7 trials per participant; mean ± SD), including very few trials on which the mouse button did not react properly).

## MEG source reconstruction

Source reconstruction was performed using Fieldtrip (*Oostenveld et al., 2011*), SPM8 (Wellcome Trust, London, United Kingdom), and the Freesurfer toolbox (*Fischl, 2012*). For each participant whole-brain T1-weighted structural magnetic resonance images (MRIs, 192 sagittal slices, 256 × 256 matrix size, 1 $mm^3$ voxel size) were acquired using a Siemens 3T Trio scanner (32-channel head coil). These were co-registered to the MEG coordinate system using a semi-automatic procedure. Individual MRIs were segmented and linearly normalized to a template brain (MNI space). Next, a volume conduction model was constructed using a single-shell model based on an 8 mm isotropic grid. We projected the sensor-level waveforms into source space using a linear constraint minimum variance (LCMV) beamformer with a regularization parameter of 7%. Then the data was collapsed onto the strongest dipole orientation based on singular value decomposition. Source reconstruction was performed on each block separately, and then concatenated for further analyses.

## Discriminant analysis

To extract neural signatures of the encoding of different variables of interest we applied a cross-validated regularized linear discriminant analysis (LDA) (*Blankertz et al., 2011*; *Parra and Sajda, 2003*) to the single trial MEG source data. LDA was applied to the data aligned to stimulus onset in 60 ms sliding windows, with 10 ms time-steps, using a spatial searchlight around each voxel consisting of the 27 neighboring voxels. For each source point $v$, the LDA identifies a projection of the multidimensional source data, $x_v(t)$, that maximally discriminates between the two conditions of interest, defined by a weight vector, $w_v$, which describes a one dimensional combination of the source data, $Y_v(t)$:

$$Y_v(t) = \sum_{i}^{27} w_v \cdot x_v(t) + c \tag{1}$$

with $i$ summing over grid points within a spatial searchlight, and $c$ being a constant. Classification performance was quantified using the area under the receiver operator characteristic (AUC) based on 6-fold cross validation. We identified clusters with significant classification performance at the group level by applying a cluster-based permutation procedure (see below). Having established a set of discriminant weights, $w_v$, one can derive single trial predictions of the neurally encoded information, $Y_v(t)$, using equation (*Equation 1*). Importantly, using cross-validation one can determine the classification weights on one set of trials, and then predict the discrimination performance for a separate trials. The value, $Y_v(t)$, of such an LDA projection can serve as a proxy to the neurally encoded signal trial information about a specific stimulus variable, and can be related for example to behavioral performance (*Grootswagers et al., 2018*; *Grootswagers et al., 2017*; *Kayser et al., 2016*; *Philiastides et al., 2014*).

## Neural en- and decoding analysis

We computed separate linear discriminant classifiers based on MEG activity in either the AV trial or the A trial, and for different to-be-classified conditions of interest. These were the location of the auditory and visual stimuli in the AV trial ($A_{AV}$, $V_{AV}$), the auditory stimulus in the A trial ($A_A$), and the responses in either trial ($R_{AV}$, $R_A$). For each stimulus location or response variable, we classified whether this variable was left- or right-lateralized. That is, we reduced the five potential stimulus locations, or the continuous response, into two conditions to derive the classifier: we grouped trials on which the respective stimulus was to the left ($-17°,–8.5°$) or right ($17°$, $8.5°$) of the fixation point, or grouped trials on which the response was to the left ($<0°$) or the right ($>0°$). The center stimuli were not used to derive the LDA classifiers. We used these classifiers in different neuro-behavioral models to elucidate neural mechanisms of the VE and VAE. We investigated models capturing potential influences of each stimulus on the neural representation of sensory information, or on the neural representation of the upcoming participant's response. In addition, we investigated models directly capturing the neuro-behavioral relation between the encoded sensory information and the response bias.

First, we determined when and where neural signatures of the encoding of single trial information, as reflected by their LDA discriminant values (c.f. *Equation 1*), were influenced by the sensory stimuli. For each searchlight and time point within a trial we determined the following models:

$$LDA_{A_{AV}} \sim 1 + \beta_{A_{AV}} \cdot A_{AV} + \beta_{V_{AV}} \cdot V_{AV}, \tag{2}$$

which captures sensory integration within the AV trials, and,

$$LDA_{A_A} \sim 1 + \beta_{A_A} \cdot A_A + \beta_{A_{AV}} \cdot A_{AV} + \beta_{V_{AV}} \cdot V_{AV}, \tag{3}$$

which captures how the encoding of the current sound in the A trial is influenced by previous audio-visual stimuli. Second, and analogously, we investigated models for the encoding of the participant's response (i.e. LDA_R_A and LDA_R_AV). It is important to note that here, the MEG activity from the stimulus- and post-stimulus period in the AV trial was used for *Equation 2*, while the MEG activity from the A trial was used for *Equation 3*.

Third, we determined the contribution of the single trial representations of acoustic and visual information to the single trial behavioral response bias with the following models:

$$VE \sim 1 + \beta_{LDA_{A\_AV}} \cdot LDA_{A_{AV}} + \beta_{LDA_{V\_AV}} \cdot LDA_{V_{AV}} \tag{4}$$

$$VAE \sim 1 + \beta_{LDA_{A\_AV}} \cdot LDA_{A_{AV}} + \beta_{LDA_{V\_AV}} \cdot LDA_{V_{AV}} \tag{5}$$

In these models, the response biases VE and VAE were the continuous localization data (in visual degrees) obtained from the participant response, while the LDA components (from *Equation 1*) reflect the continuous-valued prediction of the degree of lateralization of the respective stimulus extracted from the MEG activity. Importantly, each model is computed based on the MEG activity and the behavioral data in the respective trials (VE in the AV trial, VAE in the A trial. That is, the VE model (*Equation 4*) uses the behavioral response (VE) from the AV trial and the representation (LDA component) of the two stimuli in the MEG activity from the same trial. In contrast, the VAE model (*Equation 5*) uses the behavioral response (VAE) from the A trial, and the representation (LDA component) of the two stimuli presented in the previous AV trial, reflected in the MEG activity from the A trial. Hence, each model uses the neural representation of stimuli from a different trial. A predictor of the current sound ($A_A$) was not included in *Equation 5*, as the VAE bias quantifies the deviation of the behavioral response from the current sound (c.f. *Figure 1*).

To avoid overfitting, we computed these models based on 6-fold cross-validation, using distinct sets of trials to determine the weights of the LDA and to compute the regression models. We then computed group-level t-values for the coefficients for each regressor at each grid and time point, and assessed their significance using cluster-based permutation statistics (below). Note that for AV trials, we excluded the AV pairs with the most extreme discrepancies (±34˚), as these were inducing strong correlations between the regressors.

## Statistical analysis

To test the significance of the behavioral biases we used two-sided Wilcoxon signed rank tests, correcting for multiple tests using the Holm procedure with a family-wise error rate of p=0.05.

Group-level inference on the 3-dimensional MEG source data was obtained using randomization procedures and cluster-based statistical enhancement controlling for multiple comparisons in source space (*Maris and Oostenveld, 2007*; *Nichols and Holmes, 2002*). First, we shuffled the sign of the true single-subject effects (the signs of the chance-level corrected AUC values; the signs of single-subject regression beta's) and obtained distributions of group-level effects (means or t-values) based on 2000 randomizations. We then applied spatial clustering based on a minimal cluster size of 6 and using the sum as cluster-statistics. For testing the LDA performance, we thresholded effects based on the 99th percentile of the full-brain distribution of randomized AUC values. For testing the betas in regression models, we used parametric thresholds corresponding to a two-sided p=0.01 (tcrit = 2.81, d.f. = 23; except for the analysis for the visual location in the AV trial (tcrit = 6.60; c.f. *Figure 4*). The threshold for determining significant clusters for classification performance was p≤0.01 (two-sided), that for significant neuro-behavioral effects (*Equation 2-5*) p≤0.05 (two-sided). To simplify the statistical problem, we tested for significant spatial clusters at selected time points only. These time points were defined based on local peaks of the time courses of respective LDA AUC performance (for models in *Equations 2-3*), and the peaks for the beta time-course for the behavioral models (*Equations 4-5*). Furthermore, where possible, to test for the significance of individual regressors we applied a spatial a priori mask derived from the significance of the respective LDA AUC values to further ensure that neuro-behavioral effects originate from sources with significant encoding effects (*Giordano et al., 2017*). Note that this was not possible for $LDA_{V\_AV}$ for which we used the full brain to test for significant model effects.

## Data sharing

The behavioral data presented in *Figure 1* and LDA performance data and source regression data used to calculate the t-values in *Figures 2–5*, as well as data for *Figure 4—figure supplement 1*, have been deposited to Dryad (https://doi.org/10.5061/dryad.t0p9c93).

## Acknowledgements

This work was supported by the European Research Council (to CK ERC-2014-CoG; grant No 646657). We would like to thank Yinan Cao and Bruno Giordano for helping with the acoustic stimuli, Bruno Giordano for helpful discussions and Gavin Paterson for support with hardware and data acquisition along with Frances Crabbe.

## Additional information

### Funding

| Funder | Grant reference number | Author |
| --- | --- | --- |
| H2020 European Research Council | ERC-2014-CoG No 646657 | Christoph Kayser |

The funders had no role in study design, data collection and interpretation, or the decision to submit the work for publication.

### Author contributions

Hame Park, Conceptualization, Data curation, Formal analysis, Validation, Investigation, Visualization, Writing—original draft, Writing—review and editing; Christoph Kayser, Conceptualization, Resources, Software, Supervision, Funding acquisition, Validation, Methodology, Writing—original draft, Project administration, Writing—review and editing

### Author ORCIDs

Hame Park (ID) https://orcid.org/0000-0002-2191-2055
Christoph Kayser (ID) https://orcid.org/0000-0001-7362-5704

### Ethics

Human subjects: All participants submitted written informed consent. The study was conducted in accordance with the Declaration of Helsinki and was approved by the local ethics committee. Ethics Application No: 300140078 (College of Science and Engineering, University of Glasgow).

### Decision letter and Author response

Decision letter https://doi.org/10.7554/eLife.47001.016
Author response https://doi.org/10.7554/eLife.47001.017

## Additional files

### Supplementary files

• Transparent reporting form
DOI: https://doi.org/10.7554/eLife.47001.012

### Data availability

The behavioral data presented in Figure 1 and LDA performance data and source regression data used to calculate the t-values in Figures 2-5, as well as data for Figure 4-figure supplement 1, have been deposited to Dryad (https://doi.org/10.5061/dryad.t0p9c93).

The following dataset was generated:

| Author(s) | Year | Dataset title | Dataset URL | Database and Identifier |
| --- | --- | --- | --- | --- |
| Park H, Kayser C | 2019 | Data from: Shared neural underpinnings of multisensory integration and trial-by-trial perceptual recalibration in humans | https://doi.org/10.5061/dryad.t0p9c93 | Dryad Digital Repository, 10.5061/dryad.t0p9c93 |

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
