## [Decision Letter]

Thank you for submitting your article "Shared neural underpinnings of multisensory integration and trial-by-trial perceptual recalibration" for consideration by *eLife*. Your article has been reviewed by three peer reviewers, including Ross K Maddox as the guest Reviewing Editor and Reviewer #1, and the evaluation has been overseen by Barbara Shinn-Cunningham as the Senior Editor. The following individuals involved in review of your submission have also agreed to reveal their identity: Sarah Baum Miller (Reviewer #2) and Adam Bosen (Reviewer #3).

The reviewers have discussed the reviews with one another and the Reviewing Editor has drafted this decision to help you prepare a revised submission.

Summary:

This manuscript by Park and Kayser investigates two complementary aspects of multisensory perception: the integration of audiovisual information within a trial, as well as the impact of a multisensory event on subsequent trials. They use single trial analysis of MEG data while human participants completed a spatial localization task which probes both processes. The experimental design follows that of Wozny and Shams, 2011, which is a good method of dissociating the ventriloquism effect from the ventriloquism aftereffect. This study represents a much needed addition to the field and leverages the spatial and temporal resolution of MEG. All three reviewers viewed the manuscript positively. However, the following reviewer concerns and questions should be addressed in the revised version.

Also note that the title should provide a clear indication of the biological system under investigation. Please revise your title with this advice in mind.

Essential revisions:

The first concern relates to the time course of eye movements during the experiment. Each trial starts with a fixation period, but the Materials and methods do not specify if participants were instructed to maintain fixation during target presentation and/or localization. Eye position can bias auditory localization (Razavi et al., 2007) and the ventriloquism aftereffect is coded in a mixed eye-head reference frame (Kopco et al., 2009), so eye movements during the experiment could substantially influence the behavioral results. We would like to see a more detailed description of how eye movements were controlled or could have influenced the behavioral results. Additionally, the absence of a significant electrophysiological coding of visual location on previous trials (Figure 2A, third panel), could be a result of a shifting visual reference frame caused by eye movements. Because auditory stimuli were presented via insert tubes, the auditory stimuli would not be altered by head movements, which could explain why auditory but not visual representations were evident.

The classification is not explained clearly. In subsection “Neural en- and decoding analysis” you state that "each location was considered as a binary variable," with both left locations (-17, -8.5) collapsed, and the same done for the right locations. Was this done only for the classification outcome, or were the classification features also binarized? If they were, would this be an issue in trials where there was a VE shift but it did not cross into the other hemifield? If it was not the case, please make clear exactly what was binarized and what remained continuous.

Both the VE and VAE models (Equation 4 and 5) appear to be the same model. If this is not the case, then perhaps this could be clarified for the reader. We assume VAE would include β_LDAAn_ * LDA_An_ as in Equation 3. Please address this.

The ROC for each of the linear discriminants appears to just barely go above the chance line (Figure 2), so it seems the neural correlates of VE/VAE are very subtle. Is the small effect size more reflective of the nature of MEG signals or the nature of VE/VAE?

There seems to be a LH dominant response for β_LDAAn-1_ in the VAE and VE neural representations. Do you have any sense as to why these would be LH dominant processes, and could you comment on this? One could have (perhaps naively) assumed that any hemispheric biases would be more RH dominant in a spatial localization context.

In the MEG results, the neural locations observed to be associated with the ventriloquism effect and ventriloquism aftereffect are broadly in agreement with our expectations.

The use of generic HRTFs to simulate auditory source location, rather than presenting auditory targets from free-field speakers, requires some assumptions about how participants perceive auditory targets simulated with those HRTFs. However, given the need to electromagnetically isolate the MEG equipment and the fact that stimuli are only presented in azimuth indicates that the use of HRTFs is justified and should not alter the results substantially. If anything, the general HRTF would produce an "in the head" feeling, which may decrease the probability of fusing the auditory and visual stimuli. Given that some participants showed very little VE or VAE (Figure 2, panels B and D, individuals with means near zero), this may have occurred, but some individuals show little of either effect even with free field stimuli, so it seems that the use of generic HRTFs did not alter the expected behavioral trends. Please add justification for your use of non-individualized HRTFs and discuss any effects this may have had on your findings.

---

## [Author Response]

[…] Also note that the title should provide a clear indication of the biological system under investigation. Please revise your title with this advice in mind.

We revised the title to “Shared neural underpinnings of multisensory integration and trial-by-trial perceptual recalibration in humans”, to include the biological system investigated.

Essential revisions:The first concern relates to the time course of eye movements during the experiment. Each trial starts with a fixation period, but the Materials and methods do not specify if participants were instructed to maintain fixation during target presentation and/or localization. Eye position can bias auditory localization (Razavi et al., 2007) and the ventriloquism aftereffect is coded in a mixed eye-head reference frame (Kopco et al., 2009), so eye movements during the experiment could substantially influence the behavioral results. We would like to see a more detailed description of how eye movements were controlled or could have influenced the behavioral results.

Indeed, eye movements are a concern during spatial tasks. We had originally intended to use eye tracking for this study. However, as we placed the visual screen such as to cover a large field of view and to be close to the participants (to enhance AV binding), this became technically impossible. However, there are several reasons that make us believe that our results are not confounded by eye-movements:

First, and in line with previous studies (Wozny and Shams, 2011), we required participants to fixate a central fixation dot (Figure 1A) during the pre-stimulus, stimulus and post-stimulus intervals. Participants were generally asked to refrain from any eye-movements or blinks whenever the fixation dot was present. During the response period, i.e. while the cursor by which participants responded was displayed, participants could freely move their eyes to guide the cursor. We have extended the descriptions in the Materials and methods to clarify this:

“Importantly, participants were asked to maintain fixation during the entire pre-stimulus fixation period, the stimulus, and the post-stimulus period until the response cue appeared. During the response itself, they could freely move their eyes”.

To quantify whether eye movements were present during the pre-stimulus, stimulus and post-stimulus intervals we detected signatures of eye-movements by inspecting typical eye-related ICA components in the MEG data. Eye movement related ICA components were defined as those with frontal or frontal-lateralized topographies and time courses resembling those known for EOG artifacts (e.g. Keren et al., 2010; Hipp and Siegel, 2013) and as routinely used by our group in conjunction with MEG data, Giordano et al., 2017; Cao et al., 2019). This revealed that only 4.9% ± 1.7% (group mean ± s.e.m) of trials exhibited evidence of eye-movements, as judged from the ICA time courses. In particular, eye-movements during the stimulus presentation were rare (0.35% ± 0.61%, max 2.2%). Importantly, among those few eye-movement trials, the relevant ICA components did not exhibit the typical jump-like pattern known from changes in fixation position, and were rather suggestive of blinks (Vigário et al., 1998). This suggests that participants were indeed following the instructions and maintained fixation in the centre of the screen for the vast majority of trials. We have added these results to the manuscript (subsection: “Eye movements and the encoding of spatial information”).

Second, during the response period participants were free to move their eyes and hence eye movements were an essential part of the localization response. To avoid potential biases from the appearance of the response cursor, the starting position of this was always the centre. As a result, we expect that the relative spatial positions between fixation position, the appearance of the response cursor, and the final response given by the participant, were random.

Third, the response bias (VAE) was calculated as ‘VAE = (single trial response) – (mean of all response for this target sound location)’, as is common practice in the field (Rohe and Noppeney, 2015; Wozny and Shams, 2011). Hence, even if participants exhibited a general bias in pure auditory sound localization (e.g. the well-known central bias, Rohe and Noppeney, 2015), this would not affect our bias measure. We clarified this in the manuscript, both the Results, e.g.

“The VAE for each sound location was computed as R_A_ – mean(R_A_); whereby mean(R_A_) reflects the mean over all localization responses for this position and participant. This approach ensures that any bias in pure auditory localization does not confound the VAE effect (Rohe and Noppeney, 2015b; Wozny and Shams, 2011b).”

And the Materials and methods:

“The VAE was defined as the difference between the reported location (R_A_) minus the mean reported location for all trials of the same stimulus position. This was done to ensure that any overall bias in sound localization (e.g. a tendency to perceive sounds are closer to the midline than they actually are) would not influence this bias measure (Wozny and Shams, 2011b)”.

As the reviewers note, the precise reference frames in which audio-visual integration and the ventriloquist after-effects occurs remain unclear. Importantly, our study was not designed to pinpoint these precise reference frames, but rather to elucidate the neural mechanisms. Still, this point requires discussion, and we have devoted a new section to the Discussion (subsection: ‘The role of coordinate systems in spatial perception’). In addition, and since the reviewers pointed to two specific studies, we would like to discuss a number of issues related to these studies here. We deemed some of these details too specific to be included in the manuscript itself.

When relating our results to the two cited studies, a number of critical differences emerge. Razavi et al. report the effects of eye movements or fixation position DURING stimulus presentation AND response. In their experiments, the stimulus was presented UNTIL the participants made a response. In contrast, in the present and similar previous studies (Wozny and Shams, 2011) stimuli were brief (50 ms), in order to rule out eye movement effects during the perceptual phase. This critical difference explains, possibly, why our behavioral results differ in a number of ways from those reported in Razavi et al. In their study, the authors observe an overshoot of localization responses when participants are free to move their eyes. In our data, we observe the well-known undershoot (i.e. a central bias; see also Wozny et al., 2010, Rohe and Noppeney, 2015). This difference possibly results from a decoupling of stimulus encoding and response periods in our paradigm, while Razavi et al. effectively studied an active sound foraging behavior, in which perception and localization were continuously coupled. In another version of their experiment, Razavi et al. show that MAINTAINING a lateralized fixation position, WITHOUT moving the eyes to respond, results in a systematic spatial bias after a prolonged time period (with reported time constants varying between 9 and 52 seconds across participants). Such prolonged and systematic fixations were prohibited by our experimental paradigm, as participants reported every 4-5 seconds a new (and random) stimulus position by moving a cursor to the left (right). Hence, such prolonged maintained fixations and related effects as described in Razavi et al. cannot account for the trial-by-trial recalibration described here.

Kopčo et al. investigated spatial recalibration following prolonged adaptation to audio-visual discrepancies. Hence, again, the effect under investigation is not directly comparable to the trial-by-trial effect studied here. By dissociating fixation position and target location, they were able to show that recalibration was best explained in a mixed reference frame, composed of both head- and eye-centered coordinates. While this suggests that fixation position influences long-term recalibration, it is also worth noting that Kopčo et al. directly report that the eye position influence emerges only slowly during the prolonged adaptation (their Supplemental Figure 1, about after about 100 trials). In fact, they explicitly state (their Discussion) that initial phases of long-term adaptation emerge in head-centered coordinates. In relation to the present study this suggests that eye position effects are small at the trial-by-trial level, and short-term recalibration can be accounted for in head-centered coordinates.

Wozny, D. R., Beierholm, U. R., and Shams, L. (2010). Probability Matching as a Computational Strategy Used in Perception. PLOS Computational Biology, 6(8), e1000871. Retrieved from https://doi.org/10.1371/journal.pcbi.1000871

R. Vigário, V. Jousmäki, M. Hämäläinen, R. Hari, and E. Oja.

Independent component analysis for identification of artifacts in magnetoencephalographic recordings. In Advances in Neural Information Processing Systems 10, pages 229-235. MIT Press, 1998.

Additionally, the absence of a significant electrophysiological coding of visual location on previous trials (Figure 2A, third panel), could be a result of a shifting visual reference frame caused by eye movements. Because auditory stimuli were presented via insert tubes, the auditory stimuli would not be altered by head movements, which could explain why auditory but not visual representations were evident.

This is an interesting idea, albeit not totally trivial to conceive. As argued above, participants were fixating before, during and after the stimulus, both in the AV and A trials, while the eyes moved during the intermediate response period. Hence, even if the relevant spatial information was coded partly in eye-centered coordinates, during the relevant time period used for the MEG analysis (the post-stimulus period) participants were fixating and keeping their head still (mean head movement across participants: 2.7 ± 0.2 mm (mean ± s.e.m.)) and hence head- and eye-centered coordinates the same. As a result, a change in coordinate systems seems unlikely to underlie the rather low decoding performance for V_AV_. One possibly, however, could be that eye movements during the response period may have, in part, wiped out some of the relevant representations, as additional visual information was present (the response bar and response cursor). We have added this speculation to the Discussion:

“Furthermore, it is also possible, that eye movements during the response period may have contributed to reducing a persistent representation of visual information, in part as additional visual information was seen and processed during the response period (e.g. the response cursor).”

The classification is not explained clearly. In subsection “Neural en- and decoding analysis” you state that "each location was considered as a binary variable," with both left locations (-17, -8.5) collapsed, and the same done for the right locations. Was this done only for the classification outcome, or were the classification features also binarized? If they were, would this be an issue in trials where there was a VE shift but it did not cross into the other hemifield? If it was not the case, please make clear exactly what was binarized and what remained continuous.

The LDA analysis was designed to extract MEG activity that is sensitive to spatial information. The experiment featured five discrete spatial positions for the auditory/visual stimuli, and a continuous spatial response of the participants. To enter these spatial coordinates into a linear discriminant analysis, we grouped stimulus coordinates or participant responses to the left or right of the central midline. We have clarified the Materials and methods for how the LDA was computed as follows:

“We computed separate linear discriminant classifiers based on MEG activity in either the AV trial or the A trial, and for different to-be-classified conditions of interest. These were the location of the auditory and visual stimuli in the AV trial (A_AV_, V_AV_), the auditory stimulus in the A trial (A_A_), and the responses in either trial (R_AV_, R_A_). For each stimulus location or response variable, we classified whether this variable was left- or right-lateralized. That is, we reduced the 5 potential stimulus locations, or the continuous response, into two conditions to derive the classifier: we grouped trials on which the respective stimulus was to the left (-17, -8.5 °) or right (17, 8.5 °) of the fixation point, or grouped trials on which the response was to the left (<0°) or the right (>0°). The center stimuli were not used to derive the LDA classifiers.”

Importantly, this binarization was only used when grouping trials for deriving the LDA weights, but not for the subsequent regression analyses. We have clarified this:

“In these models, the response biases VE and VAE were the continuous localization data (in visual degrees) obtained from the participant response, while the LDA components (from Equation 1) reflect the continuous-valued prediction of the degree of lateralization of the respective stimulus extracted from the MEG activity.”

Both the VE and VAE models (Equation 4 and 5) appear to be the same model. If this is not the case, then perhaps this could be clarified for the reader. We assume VAE would include β_LDAAn_ * LDA_An_ as in Equation 3. Please address this.

Indeed, Eq. 4 and Eq. 5 may have been a bit confusing. What was missing was an explanation of which MEG activity was actually used for each model. The VE effect is studied in the AV trial, and hence the LDA is computed from the MEG activity in that trial. The VAE effect is studied in the A trial, hence the LDA is computed from MEG activity in this A trial. We now have clarified this in the descriptions of all Equations 2-5:

“Importantly, each model is computed based on the MEG activity and the behavioral data in the respective trials (VE in the AV trial, VAE in the A trial. That is, the VE model (Equation 4) uses the behavioral response (VE) from the AV trial and the representation (LDA component) of the two stimuli in the MEG activity from the same trial. In contrast, the VAE model (Equation 5) uses the behavioral response (VAE) from the A trial, and the representation (LDA component) of the two stimuli presented in the previous AV trial, reflected in the MEG activity from the A trial. Hence, each model uses the neural representation of stimuli from a different trial”

We now also explain why the regressor A_A_ was not included in Equation 5: VAE reflects the deviation of the behavioral response from the actual sound location (c.f. Figure 1) and by way of how the VAE bias was defined, the contribution of A_A_ is effectively removed. We now explicitly state this in the Results section:

“Note that because the VAE was defined relative to the average perceived location for each sound position, the actual sound position (A_A_) does not contribute to the VAE.”, as well as the Materials and methods (p26, L652-L654): “A predictor of the current sound (A_A_) was not included in Equation 5, as the VAE bias quantifies the deviation of the behavioral response from the current sound (c.f. Figure 1)”.

The ROC for each of the linear discriminants appears to just barely go above the chance line (Figure 2), so it seems the neural correlates of VE/VAE are very subtle. Is the small effect size more reflective of the nature of MEG signals or the nature of VE/VAE?

We can only offer speculative responses to this question. We report classification performance using the ROC, which is more principled than reporting e.g. the% of correctly decoded trials, as many studies do. The use of different metrics makes it difficult to compare performance values across studies. We can say that our classification method worked well, as for the AV-trial we can classify the visual target position (LDA_V_AV_; Figure 4—figure supplement 1) within the AV-trial with an ROC of about 0.9. Other studies on visual scene or objects report performance in binary classification tasks of around 60- 80% correct (e.g. Grootswagers et al., 2016; Cichy et al., 2016, Mohsenzadeh et al., 2018; Guggenmos et al., 2018). In contrast, classification performance for auditory locations is much lower in the present study. However, inspection of the literature shows that performance for classifying sound types from human EEG data using LDA (Kayser et al., 2017; Correia et al., 2015)both report an ROC of about 0.55) exhibits a similar performance. In particular, one study directly investigating the decoding of left vs. right lateralized sounds (but along the head-midline, rather than along an external fronto-parallel plane) in human EEG data reported an average performance of 75%, with some individuals only allowing only around 60% classification (Bednar et al., 2017).

This difference between classification performance for visual and acoustic stimuli remains unclear to us. It could result from a number of factors, such as i) the depth of relevant brain regions relative to the surface, and hence more noisier MEG/EEG signals; ii) more distributed representations within auditory pathways, e.g. auditory spatial maps have proven much more difficult to find than visual spatial maps (Alain et al., 2001; Alain et al., 2004; Town et al., 2017); or iii) the fact that auditory signals are simply not reflected linearly in large scale physiological signals (e.g. M/EEG), possibly because the underlying neurophysiological generators differ from those giving rise to visual evoked responses (Panzeri et al., 2015). Given the very speculative nature of this reply, we have refrained from including this in the manuscript.

Cichy, R. M., and Pantazis, D. (2016). Multivariate pattern analysis of MEG and EEG: a comparison of representational structure in time and space. *BioRxiv*, 95620. https://doi.org/10.1101/095620

Mohsenzadeh, Y., Mullin, C., Oliva, A., and Pantazis, D. (2018). The Perceptual Neural Trace of Memorable Unseen Scenes. *BioRxiv*, 414052. https://doi.org/10.1101/414052

Guggenmos, M., Sterzer, P., and Cichy, R. M. (2018). Multivariate pattern analysis for MEG: A comparison of dissimilarity measures. *NeuroImage, 173*, 434–447. https://doi.org/https://doi.org/10.1016/j.neuroimage.2018.02.044

Correia, J. M., Jansma, B., Hausfeld, L., Kikkert, S., and Bonte, M. (2015). EEG decoding of spoken words in bilingual listeners: from words to language invariant semantic-conceptual representations. *Frontiers in Psychology.*

Bednar, A., Boland, F. M., and Lalor, E. C. (2017). Different spatio-temporal electroencephalography features drive the successful decoding of binaural and monaural cues for sound localization. *European Journal of Neuroscience, 45*(5), 679–689. https://doi.org/10.1111/ejn.13524

Alain C, Arnott SR, Hevenor S, Graham S, Grady CL (2001) “What” and “where” in the human auditory system. Proc Natl Acad Sci USA 98: 12301–12306.

Arnott SR, Binns MA, Grady CL, Alain C (2004) Assessing the auditory dual-pathway model in humans. Neuroimage 22: 401–408.

Panzeri, S., Macke, J.H., Gross, J., and Kayser, C. (2015). Neural population coding: combining insights from microscopic and mass signals.. Trends in Cognitive Sciences 19(3), 162-172.

There seems to be a LH dominant response for β_LDA__An-1 in the VAE and VE neural representations. Do you have any sense as to why these would be LH dominant processes, and could you comment on this? One could have (perhaps naively) assumed that any hemispheric biases would be more RH dominant in a spatial localization context. In the MEG results, the neural locations observed to be associated with the ventriloquism effect and ventriloquism aftereffect are broadly in agreement with our expectations.

We did not have specific expectations as to the lateralization of sound encoding based on previous work. In fact, a closer look at previous studies suggests that the question about a potential lateralization of sound encoding remains largely unresolved. Clearly, sounds with lateralized azimuth generally seem to evoke stronger responses in the contralateral auditory cortex, as least when assessed using neuroimaging methods (e.g. Zierul et al., 2017; Fujiki et al., 2002). However, it is not a priori clear that such an activation difference also translates to a significant difference in the quality of neural representations, i.e. a difference in how well spatial information can be decoded from brain activity. A previous EEG study decoding sound location using a linear classifier, as we used, in fact reported bilateral contributions to sound azimuth encoding (Bednar et al., 2017). The picture seems somewhat different for sound elevation (Fujiki et al., 2002), but this is not of relevance for our paradigm. Concerning the neural correlates of multisensory recalibration the question about lateralization remains also unclear. A previous study on the long-term ventriloquist after-effect showed that the neural correlates of the perceptual effect were stronger in the left hemisphere, albeit no direct test for lateralization was performed (Zierul et al., 2017).

To directly address this question using data analysis, we subjected the respective clusters of interest (C_PAR_, C_TEMP_) to a direct tests for lateralization (Liégeois et al., 2002). Note that C_VAE_ already comprised clusters from both hemispheres, hence reflects a genuine bilateral effect. For the remaining clusters we compared classification performance (AUC) and the model β’s between the significant cluster on the left hemisphere and the corresponding site on the other hemisphere, following an approach we have already used in previous work (Giordano et al., 2017). This revealed no clear significant and systematic differences between hemispheres, suggesting that for the present data we can’t speak of statistically lateralized effects. These results are now reported in a new section in the Results (subsection: ‘Hemispheric lateralization of audio-visual integration’) and are briefly mentioned in the Discussion: “While the significant clusters were more pronounced on the left hemisphere, a direct assessment did not provide evidence for these effects to be lateralized in a strict sense (Liégeois et al., 2002)”.

Fujiki N, Riederer KA, Jousmäki V, Mäkelä JP, Hari R. (2002). Human cortical representation of virtual auditory space: differences between sound azimuth and elevation. Eur J Neurosci. 16(11):2207-13.

Bednar A, Boland F, Lalor E. (2017). Different spatio-temporal electroencephalography features drive the successful decoding of binaural and monaural cues for sound localization. Eur J Neurosci. 45(5):679-689. doi: 10.1111/ejn.13524

The use of generic HRTFs to simulate auditory source location, rather than presenting auditory targets from free-field speakers, requires some assumptions about how participants perceive auditory targets simulated with those HRTFs. However, given the need to electromagnetically isolate the MEG equipment and the fact that stimuli are only presented in azimuth indicates that the use of HRTFs is justified and should not alter the results substantially. If anything, the general HRTF would produce an "in the head" feeling, which may decrease the probability of fusing the auditory and visual stimuli. Given that some participants showed very little VE or VAE (Figure 2, panels B and D, individuals with means near zero), this may have occurred, but some individuals show little of either effect even with free field stimuli, so it seems that the use of generic HRTFs did not alter the expected behavioral trends. Please add justification for your use of non-individualized HRTFs and discuss any effects this may have had on your findings.

We had to use HRTFs (as opposed to speakers) given the circumstances in the MEG suite. We initially considered the possibility of using the pseudo-individualized HRTF with the CIPIC database. However after discussion with a colleague who actually used both the pseudo-individualized HRTFs and non-individualized HRTFs, we opted for the non-individualized HRTF (PKU-IOA HRTF database), since our priority was to bring the screen as close as possible to the participant in order to enhance co-localization, whereby we could choose different distances from the PKU-IOA database, and also we chose practical and efficient procedure, since even with the pseudo-individualized HRTFs, perfect externalization similar to real speakers will not be possible, and due to measurement errors, wrong HRTF can be chosen. We tested the sounds beforehand to make sure the participants were able to perceive the different directions of sound, and proceeded when through pilot tests, we concluded the sounds would serve its purpose in our experiment. The behavioral data demonstrate that the HRTFs produced a spatially balanced percept, i.e. there was no evidence for a specific spatial (left- or right-wards) bias in pure sound localization. As the reviewers note, the use of HRTFs may have shifted the perceived location from the external space to within-participants heads. This would effectively reduce the perceived co-localization of visual and acoustic stimuli, and hence reduce the chance to see any VE and VAE biases (Rohe and Noppeney, 2016; Wozny and Shams, 2011). However, this would only reduce the magnitude of the VAE/VE bias, but not spatially distort those. As a result, the use of non-individualized HRTFs may have added noise to the behavioral or MEG data, but should not bias the results in any specific way, as these rely on predicting the spatial direction between behavioral biases and their neural underpinnings. We have added a brief note about the use of HRTF in general as opposed to real speakers to the Materials and methods:

“The behavioral data obtained during A trials confirm that participants perceived these sounds as lateralized”

And the Discussion:

“The use of virtual sound locations, rather than e.g. an array of speakers, may have affected the participants’ tendency to bind auditory and visual cues (Fujisaki, Shimojo, Kashino, and Nishida, 2004). While the use of HRTFs is routine in neuroimaging studies on spatial localization (Rohe and Noppeney, 2015a), individual participants may perceive sounds more ‘within’ the head in contrast to these being properly externalized. While this can be a concern when determining whether audio-visual integration follows a specific (e.g. Bayes optimal) model (Meijer, Veselič, Calafiore, and Noppeney, 2019), it would not affect our results, as these are concerned with relating the trial specific bias expressed in participants behavior with the underlying neural representations. Even if visual and acoustic stimuli were not perceived as fully co-localized, this may have reduced the overall ventriloquist bias, but would not affect the neuro-behavioral correlation. Indeed, the presence of both the ventriloquist bias and the trial-by-trial recalibration effect suggests that participants were able to perceive the spatially disparate sound sources, and co-localize the sound and visual stimulus when the disparity was small.”